# The AntAWS dataset: a compilation of Antarctic automatic weather station observations

Yetang Wang[1]*,[★], Xueying Zhang[1],[★], Wentao Ning[1], Matthew A. Lazzara[2], Minghu Ding[3], Carleen H. Reijmer[4], Paul C. J. P. Smeets[4], Paolo Grigioni[5], Petra Heil[6,7], Elizabeth R. Thomas[8], David Mikolajczyk[2], Lee J. Welhouse[2], Linda M. Keller[2], Zhaosheng Zhai[1], Yuqi Sun[1], and Shugui Hou[9]*

[1]College of Geography and Environment, Shandong Normal University, Jinan 250014, China
[2]Antarctic Meteorological Research and Data Center, Space Science and Engineering Center, University of Wisconsin—Madison, Madison, Wisconsin
[3]State Key Laboratory of Severe Weather, Chinese Academy of Meteorological Sciences, Beijing 100081, China
[4]Institute for Marine and Atmospheric Research Utrecht, Utrecht University, Utrecht, Netherland
[5]Laboratory for Measurements and Observations for Environment and Climate, ENEA, 00123 Rome, Italy
[6]Australian Antarctic Division, Kingston, Tasmania, Australia
[7]Australian Antarctic Program Partnership, University of Tasmania, Hobart, Tasmania, Australia
[8] British Antarctic Survey, Cambridge, UK
[9] School of Oceanography, Shanghai Jiao Tong University, Shanghai, 200240, China

[★]These authors contributed equally to this work.

*Corresponding to: Yetang Wang (yetangwang@sdnu.edu.cn) and Shugui Hou (shuguihou@sjtu.edu.cn)

**Abstract.** A new meteorological dataset from records of Antarctic automatic weather stations (here called AntAWS dataset) at 3-hourly, daily and monthly resolutions including quality control information is presented here. This dataset integrates the measurements of air temperature, air pressure, relative humidity, and wind speed and direction from 267 AWSs available from 1980 to 2021. The AWS spatial distribution remains heterogeneous, with the majority of instruments located in near-coastal areas, and few inland on the East Antarctic Plateau. Among the 267 AWSs in total, 63 have been operating for more than 20 years, and 27 of them in excess of more than 30 years. Of the five meteorological parameters, the measurements of air temperature have the best continuity and the highest data integrity. The comprehensive compilation of AWS observations has the main aim to make them easily accessible and efficient for use in local, regional and continental studies; it may be accessed at https://doi.org/10.48567/key7-ch19 (Wang et al., 2022). This dataset is invaluable for improved characterization of the surface climatology across the Antarctic continent, to improve our understanding of Antarctic surface snow-atmosphere interactions, and to evaluate regional climate models or meteorological reanalysis products.

## 1 Introduction

Against the context of global warming, Antarctica plays an increasing role in the global sea level rise, atmospheric circulation, heat balance and climate evolution, and thus has experienced intense scientific focus (e.g., IPCC, 2019; Kennicutt et al., 2019; Rignot et al., 2019). In recent decades, much attention has been paid to changes to atmospheric variables, such as air temperature, snow accumulation, and wind

speed over the Antarctic continent (Huai et al., 2019; Dong et al., 2021; IPCC, 2021), because they have profound impact on the surface energy balance, ice sheet mass changes, as well as the ecosystem in coastal and surrounding regions (e.g., Giovinetto et al., 1990; Gregory et al., 2006; Herbei et al., 2016; Convey et al., 2018). To quantify the underlying variability and trends, accurate and continuous atmopsheric measurements are a vital prerequisite.

Extensive efforts have been made to obtain continuous atmospheric observations in Antarctica since the International Geophysical Year (IGY) in 1957/1958. For example, a total of approximately 50 staffed stations were established by the end of the IGY, of which 17 have continuous meteorological records to date (Lazzara et al., 2013; Summerhayes et al., 2008). Nevertheless, the majority of the staffed stations are concentrated along the coast, and only seven stations are located in the interior of the Antarctic continent (Allison et al., 1993), which is insufficient to resolve the atmopsheric conditions of the interior Antarctica. At the same time, harsh weather conditions and the unique geographical topography of Antarctica make it extremely difficult to install and maintain staffed weather stations. Automatic weather stations (AWSs) has the advantage of gathering meteorological data in remote areas or severe weather conditions, and help to fill the gaps of staffed weather observations (Stearns et al., 1988; Allison et al., 1993; Stearns et al., 1993; Reijmer et al., 2002; Renfrew et al., 2002). A sustained AWS network is required to observe weather and climate across the Antarctic continent (Lazzara et al., 2013).

Remote AWS became practical with the introduction of the Advanced Research and Global Observation Satellite network (ARGOS) data relay system on polar orbiting satellites in 1978, and thus real-time or near real-time meteorological data could be obtained from remote places. Based on this, numerous countries independently developed AWSs to support atmospheric observations, glaciological studies, and monitoring projects in Antarctica. In 1979, the United States Antarctic Program (USAP) supported the University of Wisconsin-Madison (UW-Madison) in the deployment of AWSs in Antarctica, mainly located in the Ross Ice Shelf and the West Antarctic Ice Sheet, beyond an initial landmark research effort by Stanford University (Stearns et al., 1993; Lazzara et al., 2012). In 1982, the Australian Antarctic Division (AAD) deployed its first AWS in Antarctica from Casey Station (Allison and Morrissy, 1983). During the International Antarctic Glaciology Program, a network from Casey Station was deployed (Allison et al., 1993). Later, the Australian National Antarctic Research Expedition (ANARE) set up an AWS network with an updated AWS version around the Lambert Glacier (Allison et al., 1998). In 1985, the Italian National Programme of Antarctic Research (PNRA) installed its first AWS, in Terra Nova Bay, named by "Mario Zucchelli". Currently its AWS network is mainly located in the Victoria Land and Antarctic Plateau. Over the Antarctic Peninsula and Dronning Maud Land, the British Antarctic Survey (BAS, who did collaborate with UW-Madison initially) and the Institute for Marine and Atmospheric Research, Utrecht University (IMAU) installed their respective AWS network. The Chinese National Antarctic Research Expedition (CHINARE) installed their PANDA AWS network, including eleven AWSs from the coast to the summit of the East Antarctic Plateau (Ding et al., 2022). There are other AWS networks in the Antarctic by the different contries (e.g., Japan, France, New Zealand, South Korea). Despite the different designs of AWSs between nations, it is common that all stations measure air temperature, air pressure, relative humidity, and wind speed and direction.

Given the funding constraints of different national Antarctic programs, AWSs provide the most economical way to gather weather data to support ongoing applications, field activities and the planning of maintenance visits. Early scientific studies based on AWSs focused on the local meteorological

processes and climatology of some basic parameters, such as temperature, pressure and wind (Stearns et al.,1993; Allison et al.,1993; Aristidi et al., 2005; Seefeldt et al., 2007). Over the Antarctic Ice Sheet (AIS), there are still missing data values at each AWS, which are a constraint for the climatological studies. Spatial and temporal interpolations are often used to fill the data gaps, and as a result, some continuous time series of meteorological elements have been created (e.g., Shuman and Stearns, 2001; Bromwich et al., 2013, 2014; Reusch and Alley, 2004). In addition, the AWS observations have also been used to evaluate and validate reanalysis products, regional climate models and remote sensing retrievals (e.g., Gallée et al., 2010; Tastula et al., 2012; Wang et al., 2013; Huai et al., 2019). Antarctic AWS observations are also used in the glaciological studies, such as estimation of snow accumulations (e.g., Wang et al., 2021), calculation of the surface energy balance (e.g., van Wessem et al., 2014), and understanding the AIS mass changes (e.g., Knuth et al., 2010).

To better characterize the regional or even continental weather and climate status over Antarctica, many attempts have been made to compile all available past and present AWS observations into the Antarctic climate database. Jacka et al. (1984) carried out the pioneering work to compile all annually and monthly averaged temperature observations of Antarctic and Southern Ocean island stations. Jones et al. (1987) assembled an integrated annual and monthly mean sea level pressure and temperature dataset from 29 weather stations located at 60°S-90°S. Stearns et al. (1993) provided a detailed description of the monthly mean climate data including monthly mean and extreme values of temperature, pressure, wind speed and direction collected and processed by the Antarctic AWSs at UW-Madison. The dataset is being continuously updated. Turner et al. (2004) described the Reference Antarctic Data for Environmental Research (READER) by the Scientific Committee on Antarctic Research (SCAR). The dataset includes the monthly and annual mean near-surface air temperature, pressure and wind speed data from 43 staffed stations and 61 AWSs. Rodrigo et al. (2013) compiled Antarctic surface wind observations from 115 AWSs to assess the performance of regional climate models, and ERA-40 and ERA-Interim reanalysis products. These AWS observation compilations generally suffer from part or all of the following limitations: the duration of datasets, single meteorological parameter, low temporal resolution, limited spatial coverage, limited or no rigorous quality control, and in some cases limited availability for the public. Most recently, Kittel compiled a near-surface weather observation database at a high temporal resolution, which to a great extent remedied the deficiency of the previous database (Kittel, 2021), and has already been used in the studies of the ice sheet surface processes, climate model validation and atmospheric diagnoses (e.g., Donat-Magnin et al., 2020; Mottram et al., 2021; Kittel et al., 2021; Kittel, 2021; Wille et al., 2021). However, these data were only qualitatively compared with models to detect and remove any outliers, and they are still not available for the public. Thus, better composition and quality control could allow for a more reliable dataset.

In this study, our main goal is to use all available records from AWSs to construct a comprehensive quality-controlled database of Antarctic meteorological parameters including air temperature, pressure, relative humidity, and wind speed and direction. The database is 3-hourly, daily, and monthly resolved. We describe the methods used to generate this dataset, including record inclusion criteria and data quality control. In addition, the main temporal and spatial features of the database are summarized.

## 2 Automatic weather station system

AWSs are ground-based meteorological data collection devices, which can run without any support all year round. All Antarctic AWSs are similar in design. They are equipped with a set of standard independent sensors, following the standards of the World Meteorological Organization (WMO, 2018). The UW-Madison AWS network at the Antarctic Meteorological Research Center (AMRC) initially consisted of dataloggers developed in-house at UW-Madison, with the AWS 2B series becoming their primary electronics system in the 1980s and early 1990s. Beginning in the late 1990s, UW-Madison switched to using commercial off-the-shelf dataloggers manufactured by Campbell Scientific. Currently, the primary AWS system used by the AMRC is consists of a Campbell Scientific CR1000 device datalogger, which is a commercial off-the-shelf system wired and programmed much like AMRC's original AWS 2B series. The CR1000 datalogger has the ability of keeping track of additional weather observations on AWSs that the AWS 2B system does not such as snow accumulation and incoming/outgoing shortwave/longwave radiation. Initially, AWSs employed by the BAS used collaborated with UW-Madison, and then switched to use the CR1000 datalogger for measurements. The IMAU Antarctic AWS Project also uses the CR1000 device and a homemade system. Most of the AWSs of the PNRA are acquisition and control units provided by Vaisala series. The glaciology program of the AAD has designed and built three different types of AWSs during the past 20 years, with the latest version being series 098 AWSs. The CHINARE AWSs consist of standard components provided by Campbell and Vaisala series, except the XFY3-1 sensor (domestic propeller anemometer) (Ding et al., 2022). The supporting framework for AWS instruments varies between models, but in general, the AWS body is made up of a mast and instrument arms fitted with different sensors. The AWS datalogger, satellite transmitter, pressure sensor, power regulating circuit and battery are generally installed in a box (or a series of boxes) at the bottom of the mast. In summer, the battery is charged by a small solar panel installed vertically near the top of the mast. However, the sensors of the AMRC AWS are mounted on Rohn tower sections, and similar towers have been used by others. Table 1 presents the different types of sensors used on the AWSs and the corresponding detailed techniques. Although the instrument manufacturers may vary across the different AWS networks, the measuring range, accuracy and resolution are identical or nearly similar. Figure 1 shows the typical AWSs in four Antarctic research projects, but other AWS may have different sensors depending on the local environment.

An AWS system can store meteorological observational data onto a datalogger, which is convenient for managing operations (e.g., DT50, CR1000, etc.). The datalogger transmits the observations through the ARGOS, carried by the National Oceanic and Atmospheric Administration (NOAA) (NOAA-19 and earlier) and Metop series of polar orbiting satellites. Figure 2 shows the data acquisition diagram of the AWSs, taking the Wisconsin AMRC AWS relay network as an example. One of the ways that AMRC receives the ARGOS' data (the archive data) is directly through file transfer protocol (FTP) services from Service ARGOS complete worldwide collection system, including all data (e.g., repeated data transmissions, etc.). These data are regularly processed into meteorological values via the quality control, and then provided to the community. AMRC also has a set of AWS units using the Iridium communications system much like this.

Each AWS measures air temperature, pressure, relative humidity and other meteorological elements within a height range of ~1 to 6 m, which are the initial height when the AWS was installed without

including the snow accumulation changes and site tilt, except for Zhongshan Station, which measures wind speed and wind direction at a height of 10 m. In fact, due to the accumulation of snow, the measurement height of each meteorological element varies over time, which may result in the notable meteorological observation disparities such as temperature and wind speed caused by the instrument height differences. Some AWS also measure air temperature, wind speed and other variables at different heights to provide near ground vertical gradient data, which is convenient to check the accuracy of data and the redundancy of certain sensors. Some AWSs have added sensors that measure snow temperature at different depths, solar radiation and snow depth, as well as a series of internal management parameters, such as voltage and internal temperature (see Fig.1).

Cost-effective AWSs provide timely research data from remote areas of Antarctica throughout the year. Maintenance is still needed, and generally one visit is performed per summer to ensure that electric power generation and battery capacity are sufficient for polar night operation. However, several AWSs are not revisited after initial deployment. For example, since its first deployment in October 1984, AWS GC41 has been operating continuously in the interior of Antarctica with no maintenance access. The accuracy of the data from these sites can only be estimated by the internal consistency of the diverse sensors.

Table 1. The sensor types used on the automatic weather stations and the technical specifications.

| Institution | Sensor | Type | Specifications | | |
|---|---|---|---|---|---|
| | | | Range | Resolution | Accuracy |
| AMRC | Air temperature | Weed PRT Two-wire bridge | to -100°C minimum | 0.125°C | ±0.5°C |
| | | RM Young 43347 RTD 1000-ohm PRT | to -100°C minimum | 0.01°C | ±0.3°C |
| | | Apogee ST-110 Thermistor | to -100°C minimum | 0.01°C | 0.1°C above 0°C 0.15°C below 0°C |
| | Relative humidity | Vaisala HMP14UT | 0 to 100% | 0.04% | ±4% |
| | | Vaisala HMP31UT Vaisala HMP35A/D Vaisala HMP45A/D | 0 to 100% | 0.04% | ±2% above -20°C |
| | | Vaisala HMP155 | 0 to 100% | 0.04% | ±2% above -40°C ±5% above -40° to -60°C |
| | Air pressure | Paroscientific Model 215 A | 0 to 1100 hPa | 0.04 hPa | ±0.1 hPa |
| | | CSI 105/PTB101 | 0 to 1100 hPa | 0.1 hPa | ±3 hPa |
| | | CSI 106/PTB110 | 500 to 1100 hPa | 0.1 hPa | ±1.5hPa |
| | Wind speed | Bendix Model 120 Aerovane | 0 to 60 m s$^{-1}$ | 0.25 m s$^{-1}$ | ±0.5 m s-1 |

| | | | | | |
|---|---|---|---|---|---|
| | | Belfort Model 122/123 | | | |
| | | RM Young 05103/106 | 0 to 60 m s$^{-1}$ | 0.2 m s$^{-1}$ | ±0.3 m s$^{-1}$ |
| | | Taylor Model 201 High Wind System | 0 to 60 m s$^{-1}$ | 0.33 m s$^{-1}$ | ±2 m s$^{-1}$ |
| | Wind direction | Bendix Model 120 Aerovane Belfort Model 122/123 RM Young 05103/106/ Taylor Model 201 High Wind System | 0 to 360° | 1.5° | ±3° |
| PNRA | Air temperature | Vaisala HMP45C/D | -40 to +60°C | - | ±0.2°C |
| | | Vaisala HMP155 | to -80°C minimum | - | (0.2260-0028*Ta) °C |
| | Relative humidity | Vaisala HMP45D | 0 to 100% | 0.04% | ±2% above -20°C |
| | | Vaisala HMP155 | 0 to 100% | 0.04% | ±2% above -40°C ±5% above -40° to -60°C |
| | Air pressure | CS106 Barometer | 500 to 1100 hPa | 0.1 hPa | ±1.5 hPa (-40 to +60°C) |
| | | BARO1 | 500 to 1100 hPa | 0.01 hPa | ±0.15 hPa (-40 to +60°C) |
| | | PTB200 | 600 to 1100 hPa | 0.01 hPa | ±0.15 hPa (-40 to +60°C) |
| | Wind speed | Vaisala WAA151 | 0.4 to 75 m s$^{-1}$ | - | ±0.5 m s$^{-1}$ |
| | | RM Young 05103/106 | 0 to 60 m s-1 | 0.2 m s-1 | ±0.3 m s-1 |
| | Wind direction | Vaisala WAV151 | 0 to 360° | 2.8° | ±3° |
| | | RM Young 05103 | 0 to 360° | 1.5° | ±3° |
| IMAU | Air temperature | Vaisala HMP35AC | -80 to +56°C | - | ±0.3°C |
| | Relative humidity | Vaisala HMP35AC | 0 to 100% | - | ±2% (RH<90%) ±3% (RH>90%) |
| | Air pressure | Vaisala PTB101B | 600 to 1060 hPa | - | ±4 hPa |
| | Wind speed | RM Young 05103 | 0 to 60 m s$^{-1}$ | 0.2 m s-1 | ±0.3 m s-1 |
| | Wind direction | RM Young 05103 | 0 to 360° | 1.5° | ±3° |
| AAD | Air temperature | FS23D thermistor in ratiometric circuit | -99 to +13°C | 0.02°C | ±0.05°C |
| | Relative humidity | Vaisala HMP35D | 0 to 100% | 2% | ±2% (RH<90%) ±3% (RH>90%) |
| | Air pressure | Paroscientific Digiquartz 6015A | 0 to 1100 hPa; | 0.1 hPa | ±0.2 hPa |

| | | | | | |
|---|---|---|---|---|---|
| | | | DomeA: 530 to 610 hPa; Eagle: 635 to 735 hPa; LGB69: 691 to 791 hPa | | |
| | Wind speed | 3-cup anemometer with R M Young 12170C cup set, and AAD built body and mechanism | 0 to 51 m s$^{-1}$ | 0.1 m s$^{-1}$ | ±0.5 m s$^{-1}$ |
| | Wind direction | Aanderaa 3590B wind vane Aanderaa 2750 | 0 to 360° | 6° | ±6° |
| BAS | Air temperature | CSI RTD 100-ohm PRT | to -100°C minimum | 0.01°C | ±0.5°C |
| | | Weed PRT Two-wire bridge | to -100°C minimum | 0.125°C | ±0.5°C |
| | | Vaisala HMP35D/45D | -40 to +60°C | - | ±0.2°C |
| | Relative humidity | Vaisala HMP35D | 0 to 100% | 2% | ±2% (RH<90%) ±3% (RH>90%) |
| | | Vaisala HMP45A/D | 0 to 100% | 0.04% | ±2% above -20°C |
| | Air pressure | Paroscientific Model 215 A | 0 to 1100 hPa | 0.04 hPa | ±0.1 hPa |
| | Wind speed | Propeller-vane anemometer | - | - | - |
| | | Belfort Model 122/123 | 0 to 60 m s$^{-1}$ | 0.25 m s$^{-1}$ | ±0.5 m s$^{-1}$ |
| | | RM Young 05103/106 | 0 to 60 m s$^{-1}$ | 0.2 m s$^{-1}$ | ±0.3 m s$^{-1}$ |
| | Wind direction | Propeller-vane anemometer | - | - | - |
| | | Belfort Model 122/123 RM Young 05103/106 | 0 to 360° | 1.5° | ±3° |
| CHINARE | Air temperature | HMP155 resistance probe | to -80°C minimum | - | (0.2260-0.0028*Ta) °C |
| | | Campbell 109 | - | - | (0.2260-0.0028*Ta) °C |
| | | FS23D thermistors | -99 to +13°C | 0.02°C | ±0.05°C |
| | | Weed PRT Two-wire bridge | to -100°C minimum | 0.125°C | ±0.5°C |
| | Relative humidity | HMP155 | 0 to 100% | 0.04% | ±2% above -40°C ±5% above -40° to -60°C |

| | | HMP35A/D | 0 to 100% | 0.04% | ±2% above -20°C |
|---|---|---|---|---|---|
| | Air pressure | CS106 Barometer | 500 to 1100 hPa | 0.1 hPa | ±1.5 hPa (-40 to +60°C) |
| | | PTB110 | 0 to 1100 hPa | 0.1 hPa | ±1.5hPa |
| | | PTB210 | - | - | ±0.5hPa |
| | | 6015A | 0 to 1100 hPa; DomeA: 530 to 610 hPa; Eagle: 635 to 735 hPa | 0.1 hPa | ±0.2 hPa |
| | | Paroscientific Model 215 A | 0 to 1100 hPa | 0.04 hPa | ±0.1 hPa |
| | Wind speed | XFY3-1 | 0.3 to 50 m s$^{-1}$ | - | ±1m s$^{-1}$ |
| | | 12170C | 0 to 51 m s$^{-1}$ | 0.1 m s$^{-1}$ | ±0.5 m s$^{-1}$ |
| | | RMYoung | 0 to 60 m s$^{-1}$ | 0.2 m s$^{-1}$ | ±0.5 m s$^{-1}$ |
| | Wind direction | XFY3-1 | 0 to 360° | - | ±5° |
| | | 10K Ohmpot | 0 to 355° | 1.5° | ±3° |
| | | 3590B | 0 to 360° | 6° | ±6° |
| POLENET | Air temperature | Vaisala WXT520 | - | - | ±3°C |
| | Relative humidity | Vaisala WXT520 | - | - | ±3% |
| | Air pressure | Vaisala WXT520 | - | - | ±3 hPa |
| | Wind speed | - | - | - | - |
| | Wind direction | - | - | - | - |

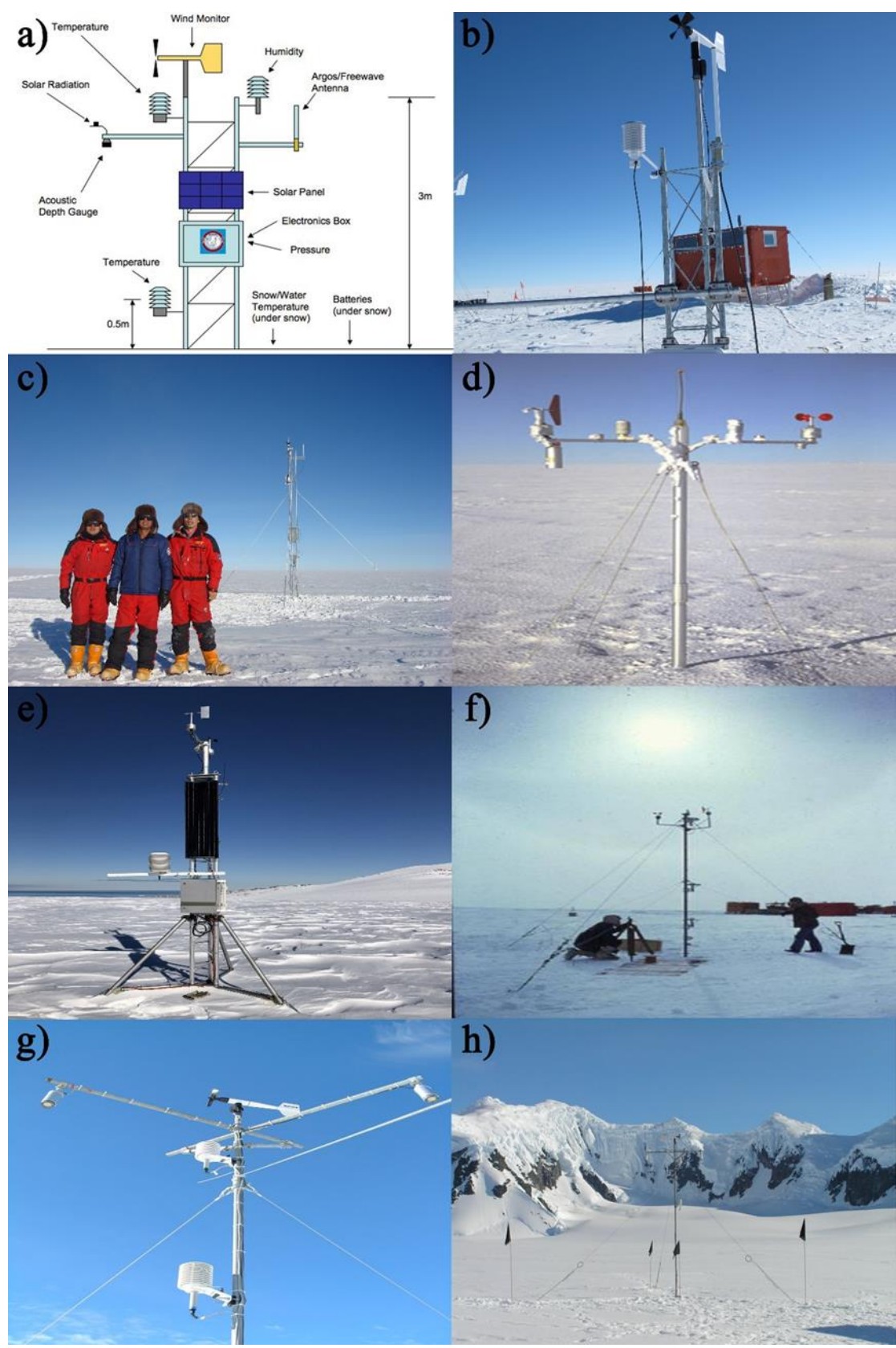

Fig.1. Typical AWSs of the six research institutions, but the sensors at other sites vary slightly
depending on the local environment. a) AMRC-CR1000 device, b) AMRC- AGO-4, c) AMRC and
CHINARE-Panda_South, d) IMAU-AWS10, e) PNRA-Maria, f) AAD-LGB00, g) BAS-the sensors
used on Latady, h) BAS-Latady.
a) http://amrc.ssec.wisc.edu/news/2010-May-01.html
b) https://amrc.ssec.wisc.edu/aws/images/station_images/AGO_4.jpg
c) personal communication with Minghu Ding.
185 d) https://www.projects.science.uu.nl/iceclimate/aws/technical.php
e) https://www.climantartide.it/attivita/aws/index.php?lang=en
f) personal communication with Ian Allison
188 g) and h) https://ramadda.data.bas.ac.uk/repository/entry/show?entryid=synth%3A44d1a477-0852-
189 4620-a1f4-63f559b44e94%3AL0RvY3VtZW50cy9waG90b3NfYXdz

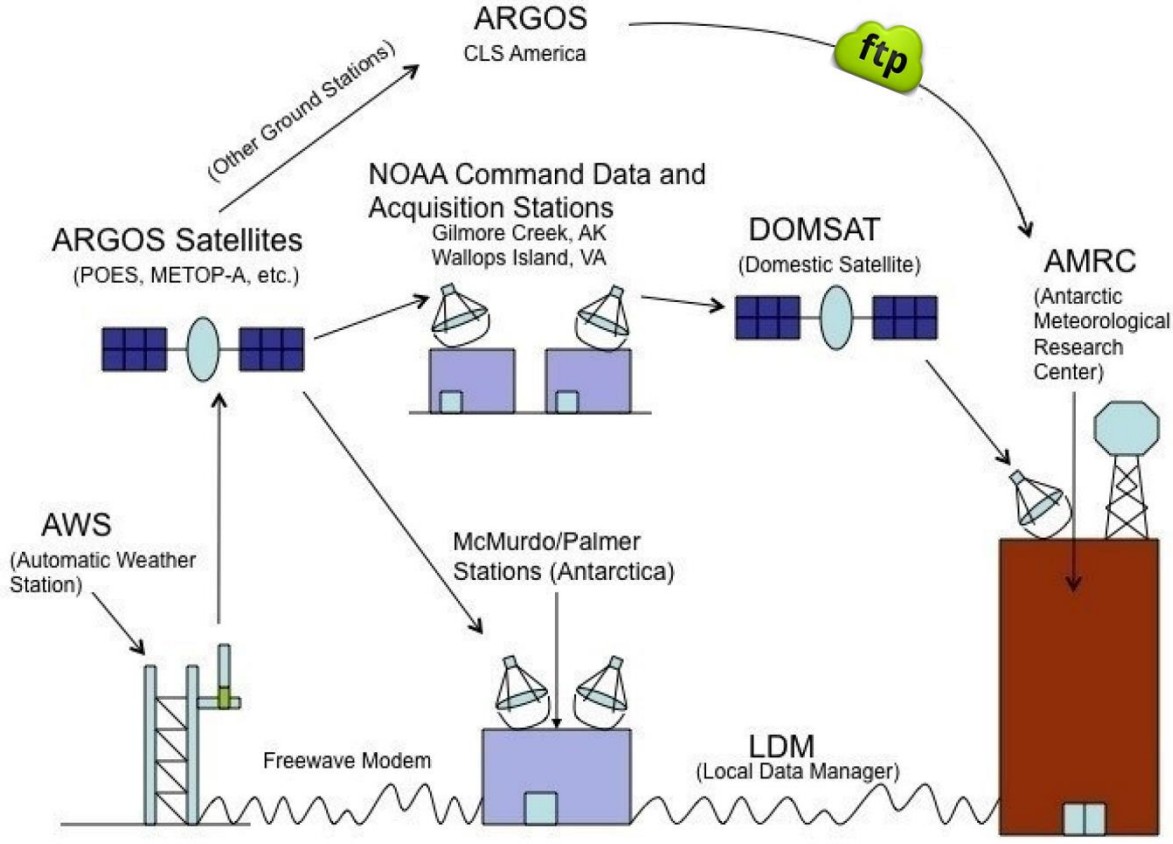

Fig.2. Data acquisition diagram of AWS, using AMRC as an example.
http://amrc.ssec.wisc.edu/aws/images/datastream_v2.jpg

## 3 Data processing

### 3.1 Data collections and sources

The AWS meteorological observations were obtained from seven Antarctic AWS project databases, including the CHINARE (https://doi.org/10.11888/Atmos.tpdc.272721), the BAS (https://data.bas.ac.uk/datasets.php), the PNRA (http://www.climantartide.it), the IMAU Antarctic AWS Project (https://www.projects.science.uu.nl/iceclimate/aws/antarctica.php) (data available from https://doi.org/10.1594/PANGAEA.910473), the AAD (http://aws.cdaso.cloud.edu.au/datapage.html), the AMRC (http://amrc.ssec.wisc.edu/) at the University of Wisconsin (Lazzara et al., 2012), and the Polar Earth Observing Network (POLENET) program (https://www.unavco.org/). The AMRC includes not only its own AWS network but also brings together data from several Antarctic research programs, such as the Japanese Antarctic Research Expedition (JARE), the French Antarctic Program (Institut Polaire Francais-Paul Emile Victor, IPEV), the AAD, the BAS and the CHINARE. The JARE installed and maintained JASE2007, Dome Fuji, Mizuho and Relay Station on the East Antarctic Plateau. The IPEV installed and took charge of the AWSs from the Adélie Coast to Dome C, including Port Martin, D-10, D-17, D-47, D-85, Dome C and Dome C II. Cape Denison AWS on the Adélie Coast is serviced by the AAD. The BAS installed and maintained the AWSs on the Antarctic Peninsula and the East Antarctic Plateau, including Butler Island, Larsen Ice Shelf, Limbert, Sky-Blu, Fossil Bluff, Dismal Island and Baldrick. The PANDA-South AWS, located on the East Antarctic Plateau, is a cooperation between CHINARE and AMRC, which was installed, maintained and operated by CHINARE.

First, we excluded AWS with data coverage of less than one year. Then, all available records from the remaining stations were collected. Finally, measurements from 267 AWSs were compiled, including at least one of the five meteorological variables: near surface air temperature, relative humidity, air pressure, wind speed and wind direction. Fig.3 shows the spatial distribution of the 267 AWSs, and the corresponding longitude and latitude coordinates, elevation and data sources of these AWSs are summarized in Table S1.

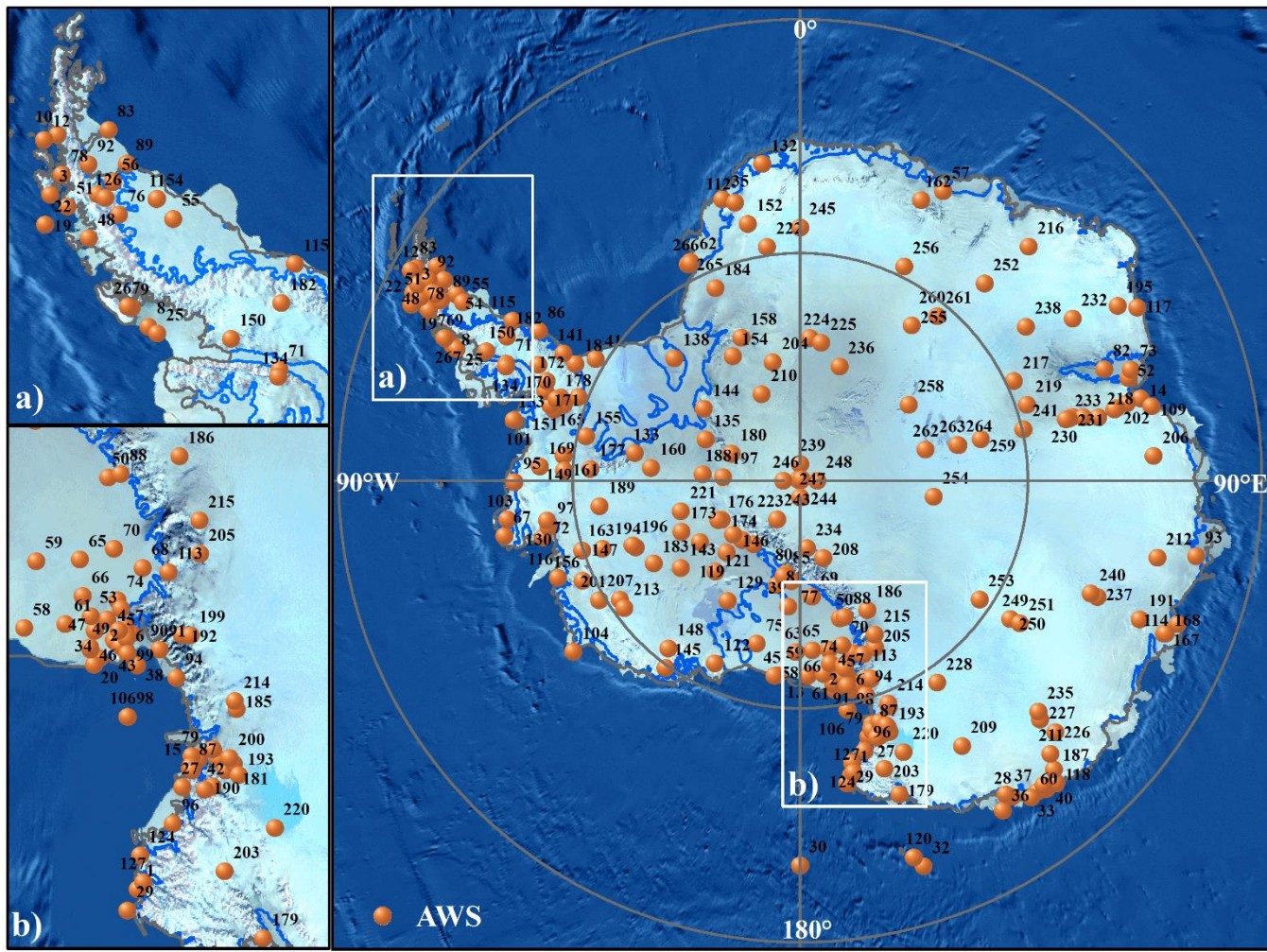

Fig.3. Map of the 267 Automatic Weather Stations (AWSs) in this study, where the numbers (1-267) correspond to NO. in Table S1.

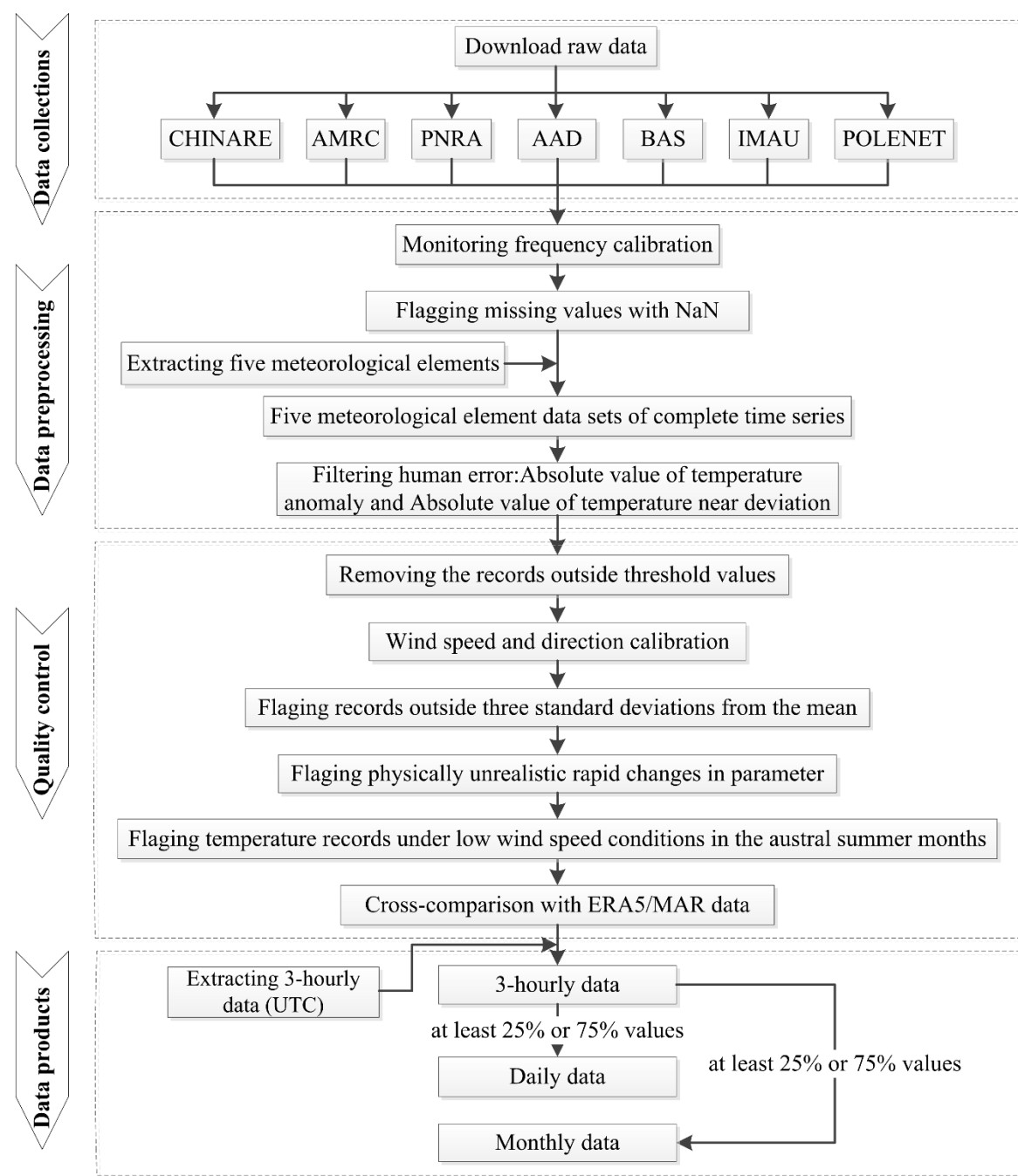

Fig.4. Description of AWSs data processing process.

**3.2 Quality control**

The quality check of observational data is aimed at detecting missing data and errors to provide the highest
possible standard of accuracy. Our compilation is based on the hourly and 3-hourly synoptic

measurements from AWSs, which were subjected to quality checks by data providers, including: a coarse error check using threshold values at the time of decoding; manually filtering errors or gaps due to the presence of instrument failures such as sensor freezing and screen covered by snow/frost; transmissions issues through the datalogger, Global Telecommunications System (GTS) or ARGOS; and changes in units. Despite the quality checks, previous studies pointed out that cautions should be made on using these AWS data, at least wind speed data, which are the least reliable variable of the measurements (e.g., Stearns et al.,1993). To perform a more rigorous quality control, a set of interactive quality control programs using interactive data language (IDL) software was developed for the quality check of the AMRC data (Lazzara et al., 2012). In our compilation, we use the 3-hourly AMRC AWS data through preliminary quality control. Since our objective was to construct a dataset with high quality, restrictive quality control criteria were used to filter the compiled data from a variety of sources.

First, we removed the records from the dataset outside the measurement range of sensors installed over the AWSs (Table 2). Data with zero values for both wind speed and direction were also eliminated. Furthermore, if the wind speed and direction values remained unchanged for 6 consecutive hours which are likely caused by sensor freezing, the values were set to the null values (NA). Secondly, the mean and standard deviation were calculated for the 3-hourly data in each month. We also checked physically unrealistic rapid synoptic variability in the parameters using the 6-h change threshold values of 10 hPa for surface pressure, 5°C for air temperature, 40 kt for wind speed (Turner et al., 2004). Following Lazzara et al. (2012), the observation values exceeding three standard deviations from the mean were considered to be possibly erroneous, and thus were flagged. Thirdly, we flagged the air temperature records in the austral summer months (December-January-February) during the low-wind speed conditions (less than $2 \text{ m s}^{-1}$), which can result in a warm temperature bias during this period because of the lack of ventilation (Genthon et al., 2011; Lazzara et al., 2012; Jones et al., 2016). At last, after these physically-based filters, we performed a visual cross-comparison of each time series of the filtered data with the corresponding outputs of ERA-5 (Hersbach et al., 2020) and MAR (Kittel, 2021), to further remove outliers and improve the reliability of the dataset.

Table 2. Threshold values used in the quality control process for each measured variable.

| Variable | Units | Low threshold | High threshold |
|---|---|---|---|
| Temperature | °C | -100 | 15 |
| Pressure | hPa | 0 | 1100 |
| Wind Speed | m/s | 0 | 60 |
| Wind Direction | ° | 0 | 360 |
| Relative Humidity | % | 0 | 100 |

## 3.3 Averaging procedure

For all meteorological parameters, daily and monthly mean values are calculated by the 3-hourly data (eight values a day, between 00:00 and 21:00 UTC). Unfortunately, a number of occurred events may result in data gaps because of only checked periodically. For daily values to be included, at least two 3-hourly observed values (25%) must be available in that day, since less than 25% of the 3-hourly observations do not well capture the weather conditions of a day, and a good daily average cannot be

obtained. Then, if at least 25% of the 3-hourly observations are available in a month, we calculate a
monthly average. For monthly data, when less than 25% of the 3-hourly observations are available, this
value typically occurs when a station starts or stops during the month. This may lead to the deviation of
the monthly average, especially in the period of rapid changes in meteorological elements such as air
temperature. All missing values are set to NA. To provide more reliable daily and monthly values, we
also calculate the daily and monthly products using a 75% threshold, that is, at least six 3-hourly observed
values are available, as in Kittel (2021).

**4 Description of the AntAWS dataset**

**4.1 Air temperature**

Air temperature is a sensitive indicator of the climate extremes experienced by the whole continent. It is
measured at the heights of approximately 3 m above the ground, using a thermistor (such as Apogee ST-
110 Thermistor and FS23D thermistor in ratiometric circuit) or resistive platinum probe (such as PRT
series and Vaisala HMP series). The air temperature sensor is installed in the AWS' naturally ventilated
radiation shields to protect the sensor from direct sunlight, and the measurement uncertainty is within
±0.5℃. It should be emphasized that over areas with strong temperature inversions, especially the
Antarctic Plateau in winter, near-surface air temperature is influenced by changes in the height of sensors
installed on the AWS (generally a relative "lowering") caused by snow accumulation (Genthon et al.,
2021).
Figure.5 and Table S2, Table S3, Table S4 show the mean, maximum and minimum values of 3-hourly,
daily, and monthly air temperature from each AWS. The overall statistical results show the effects of sea-
land distribution and elevation, as the air temperature in coastal areas is generally higher than that in
inland areas, showing a gradual decrease from coastal to inland areas. The near-surface temperature is
evidently affected by elevation and decreases significantly with the elevation increase (Fig.5). Fig.6 and
Table S2 show that the mean temperature of 3-hourly data ranges from -59.94 ℃ to 2.13 ℃. The extreme
maximum temperatures of the Antarctic Peninsula, most of the West AIS, Ross Ice Shelf and Victoria
Land are almost all over 0℃. The warmest AWSs are South Georgia 1, South Georgia 3 and King Edward
Pt, with the elevations of 85 m, 53 m and 346 m respectively, and the maximum temperature can reach
15℃. The AWSs with extreme minimum temperatures below -70℃ are mainly distributed in the East
Antarctic Plateau. The minimum temperature value is lower than -82 ℃, occurring at aws12, aws13,
Dome C and Dome F. Statistics of the daily air temperature indicate that the daily mean air temperature
values range from -58.42 ℃ to 2.36 ℃ (Table S3). The maximum daily temperature occurs at King
Edward Pt station on the Berkner Island, reaching 13.95 ℃. The lowest daily temperature is -83.51℃ at
aws13. According to the statistical results of monthly data in Table S4, the mean temperature of monthly
data ranges from -59.02℃ to 2.32 ℃. The King Edward Pt station still has the highest monthly averaged
air temperature of 5.9 ℃. Concordia, located on the East Antarctic Plateau, has the lowest monthly
temperature of -71.76℃.

## 4.2 Air pressure

All the AAD AWSs use Paroscientific digiquartz barometers, with an accuracy of ±0.2 hPa and a resolution of 0.1 hPa. AMRC AWSs also use Paroscientific digiquartz barometers (Paroscientific Model 215 A), which have a higher resolution of 0.04 hPa and accuracy of ±0.1 hPa. Most AWSs at other institutions use Vaisala's PTB series and Campbell's CS series. Both series of barometers use Vaisala's BAROCAP silicon capacitive absolute pressure sensor, which have excellent accuracy, repeatability, and long-term stability over a wide range of operating temperatures. The barometer, kept in the electronics enclosure measures, the station pressure and is not corrected to sea level. The accuracy of all air pressure measurements ranges from 0.15 hPa to 4 hPa, depending on the sensor used.

Fig.6 and Table S2 show the mean, maximum and minimum pressure of the 267 AWSs at 3-hourly time resolution. The range of the mean air pressure values goes between 573.49 hPa and 996.24 hPa. AWSs with 3-hourly average pressure greater than 900 hPa are mainly located along the coast of the Ross Ice Shelf, Antarctic Peninsula, Dronning Maud Land, the Lambert Glacier Basin, and Victoria Land. The maximum 3-hourly air pressure is 1039.2 hPa at South Georgia 3, followed by the station on the Larsen Ice Shelf of the Antarctic Peninsula. The minimum (536 hPa) is present at Dome A station, with an elevation of 4093 m. Mainly affected by elevation, the mean, maximum and minimum air pressure decreases with the increase of altitude and spatially decreases from the coast to the interior (Fig. 5). The major features of the spatial distribution of daily and monthly air pressure are almost the same as those of 3-hourly data.

## 4.3 Relative humidity

The height of the humidity sensor is often the same as that of the air temperature probe. Correct measurements of relative humidity are key to calculate sublimation. However, it is quite difficult to accurately measure, especially in Antarctica. The original network did not include such measurements, but humidity detectors (Vaisala HMP Series) have been deployed since about 1990. Humidity measurements are based on a capacitive thin film polymer sensor. The resolution of the series of humidity sensors is approximately 1%, and the annual drift in the field is approximately ±2 ~ 3%. The Vaisala humicap, which itself takes the conversion of ice and water form into account, is factory calibrated to provide RH with respect to liquid water even at below-freezing temperatures (Amory, 2020; Genthon, et al., 2013). The relative humidity is computed with respect to liquid water. Data should be converted to get RH with respect to ice using the method of Goff and Gratch (1945) (Amory, 2020), but these additional computed data are left for forthcoming papers. In Antarctica, even near the surface, the relative humidity with respect to ice often reaches well over 100%, and this is especially frequent on the high Antarctic plateau where supersaturation often occurs (Genthon et al., 2017, 2022). The sensors used on the AWS cannot report supersaturation and measure humility above 100%, and as a consquency the humidity data are biased low there.

Many AWSs lack relative humidity measurements in consecutive years or entirely, which brings great challenges to humidity research over the whole Antarctic continent. The relative humidity of the coastal AWSs is usually higher than that of the inland AWSs, and shows similar spatial patterns with air temperature.

## 4.4 Wind speeds and directions

Wind speeds and directions are monitored at a height of approximately 3 m above the ice sheet surface (Lazzara et al., 2012). It is notable that at the Zhongshan station, the 10 m wind directions are measured. Due to the influence of katabatic wind, the wind directions at this station are relatively stable and resemble the 3m wind directions (Ma et al., 2014). Different sensors are used to measure wind speed and direction at different AWSs. The most widely used model is R. M. Young Company 05103/106, in which wind speeds are measured using an impeller anemometer that is a helical, four-blade impeller. The rotation of the impeller generates a signal proportional to wind speeds, and wind directions are measured using a potentiometer. In addition, some AWSs adopt the heated Vaisala WA15 series, which is based on precise sensors mounted on cross arms. Its WAA151 anemometer has the characteristics of fast response and low threshold. Similarly, the optoelectronic vane-WAV151 has the advantages of counterbalance, sensitivity, accuracy and low threshold. It is more suitable for more demanding wind measurements. The measurement accuracy of wind speeds is approximately $\pm 0.5$ m s$^{-1}$, and wind direction is $\pm 3°$. The wind direction listed is clockwise from $0°$ to $360°$ (so $90°$ are east, $180°$ are south, and $270°$ are west). The stations established by CHINARE use a domestic propeller anemometer (XFY3-1 sensor), which can measure the wind speed and direction of horizontal airflow at very low critical wind speed, with an uncertainty of $\pm 1$ m s$^{-1}$ and $\pm 5°$, respectively (Ding et al., 2022). It is important to recall that wind speed varies strongly with height in the first few meters above the surface, and the height of the sensors above surface gradually decreases with snow accumulation, causing poorly known variations of the instrument height above the snow surface and affects the data quality and consistency (Genthon et al., 2021). Still, information on the evolution of wind speed with time is important, but the modulus is not well known and not consistent in the dataset. To improve the accuracy of air temperature and wind observations, the vertical temperature and wind profiles should be corrected by accounting for the sensor height variations, as done by Ma et al. (2008) and Smeets et al. (2018). However, this additional computed data will be left until we have sufficient snow height data.

The results of Fig.6 and Table S2, 3 and 4 show that wind speed is consistent whether parsed in 3 hourly values or in daily and monthly values, and so is wind direction. The mean near-surface wind speeds of the 267 AWSs vary from 2.17 to 23.66 m s$^{-1}$. The average wind speed is higher along the East AIS coast, where the average wind speed exceeds 20 m s$^{-1}$ (e.g., Cape Denison, Lucia, Virginia and Zoraida stations). The average wind speed at AGO-5, Dome C, Dome F, and Dome A stations on the Antarctic inland plateau is less than 3 m s$^{-1}$, mainly due to the gentler surface slopes of the inland plateau (Van den Broeke and Van Lipzig, 2003). The maximum wind speed (exceeding 60 m s$^{-1}$) is observed at Alessandra, Eneide, Lanyon, Lola, Lucia, Minna Bluff, Rita, Silvia, Sofia, Sofiab, Virginia and Zoraida stations in North Victoria Land. Spatial patterns of wind speed are generally high along the coast and low on the inland ice sheet, which is mainly determined by the terrain and pressure gradient from coastal to inland. Southerly or easterly winds prevail over most of the AIS, influenced by circumpolar westerly winds, katabatic winds, large-scale pressure gradient forces and topography, which contributes to drive the movement of the AIS atmospheric boundary layer (Van den Broeke et al., 2002). The winds over the AIS are persistent throughout most of the year, which is reflected in a high mean value of daily mean constancy of the wind direction (defined as the ratio of the magnitude of the mean wind vector to the scalar average wind speed) ($\geq 0.6$) for the majority of the AWSs (Fig.6).

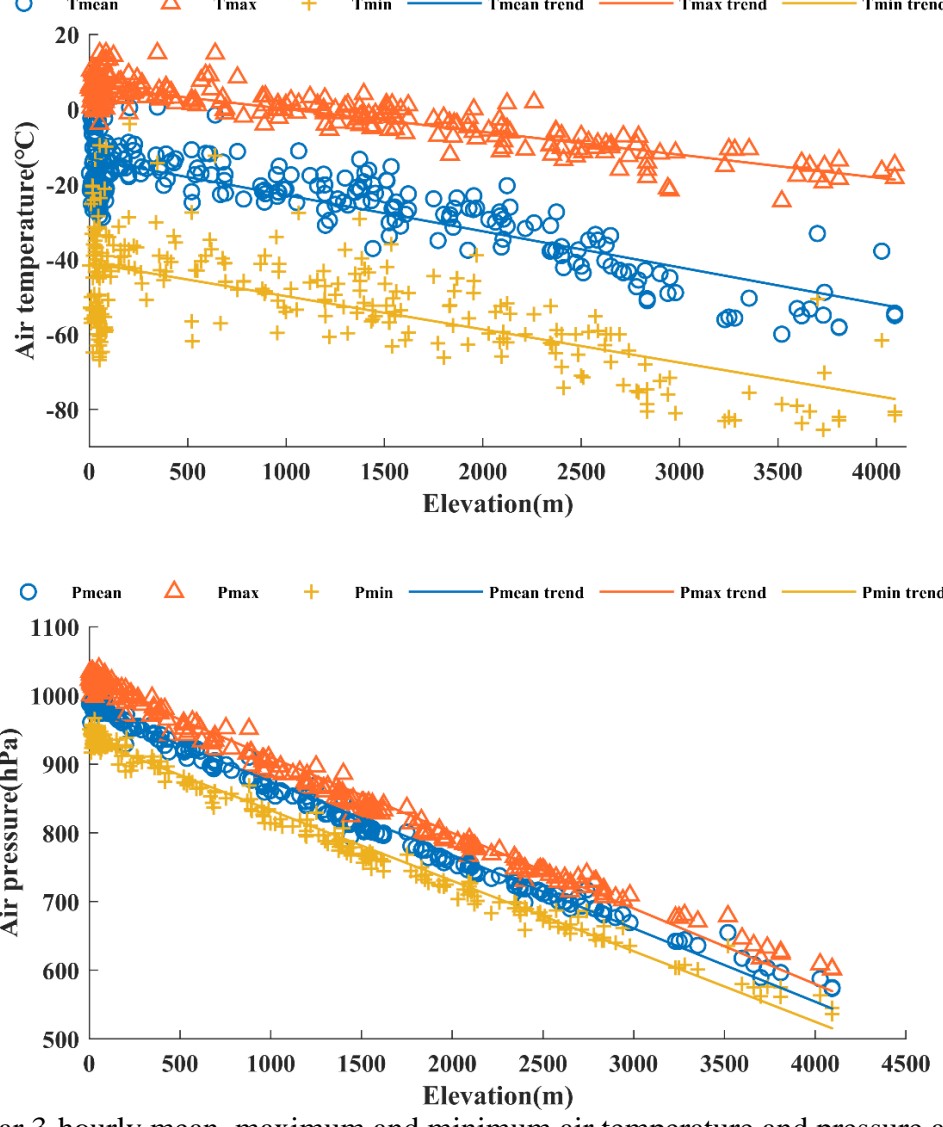

Fig.5. Multiyear 3-hourly mean, maximum and minimum air temperature and pressure as a function of
elevation.

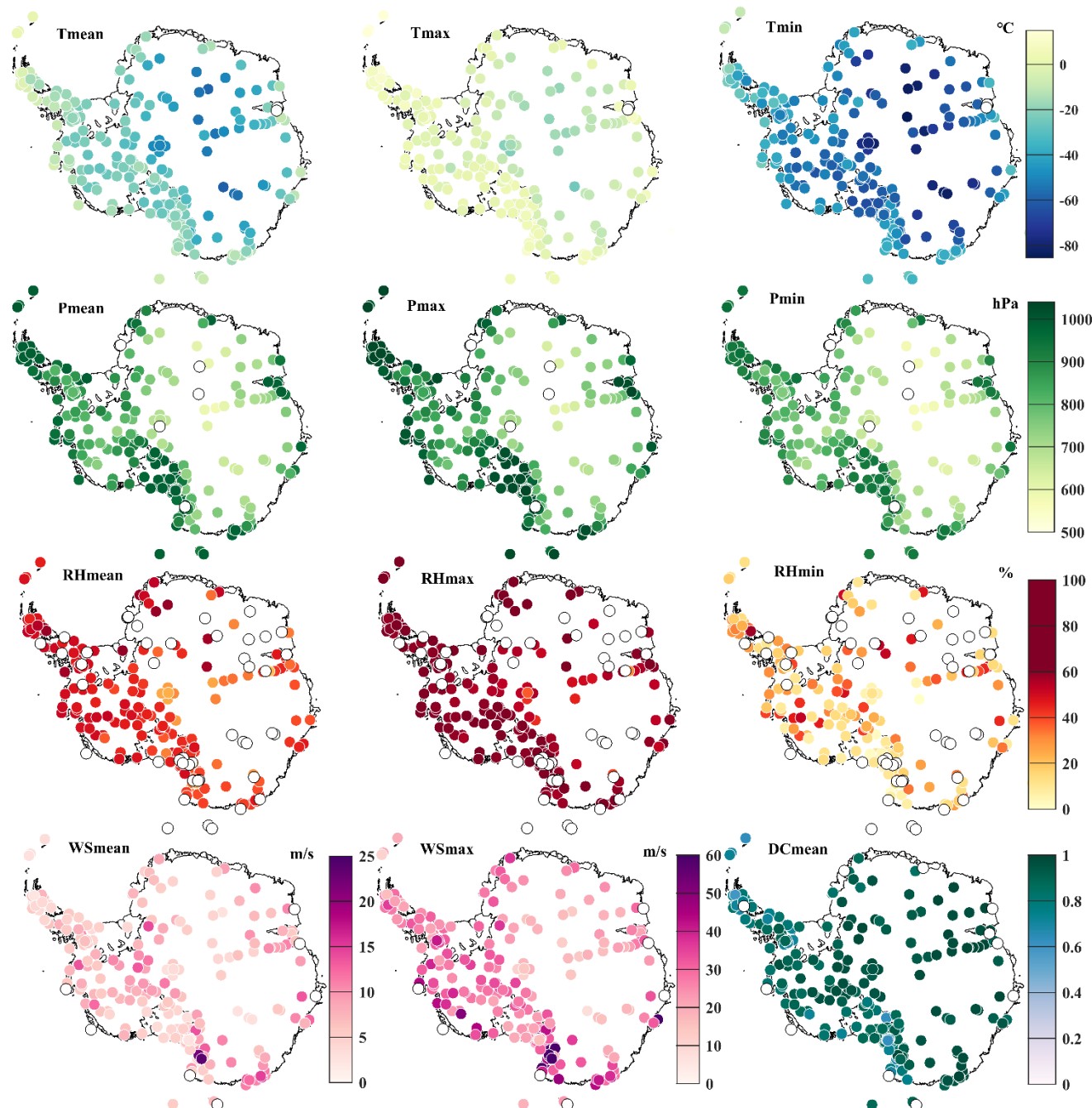

Fig.6. Spatial distribution of AWS' multiyear 3-hourly mean, maximum and minimum meteorological
elements (temperature, pressure, relative humidity, wind speed) and daily mean constancy of the wind
direction (DC) during 1980-2021. White circles represent missing data. Tmean is mean temperature,
Tmax means maximum temperature, Tmin is minmum temperature, Pmean is mean pressure, Pmax is
maximum pressure, Pmin is minmum pressure, RHmean is mean relative humidity, RHmax is maximum

relative humidity, RHmin is minmum relative humidity, WSmean is mean wind speed, WSmax is
maximum wind speed, and DCmean is daily mean constancy of the wind direction.

**5 Spatiotemporal characteristics of the AntAWS dataset**

**5.1 Spatial coverage of AWS records**

The spatial distribution of AWSs is heterogeneous over the AIS. On the whole, since 1980, the number
and coverage of AWSs have been gradually increasing (see Fig.7, Fig.8, and Table S5). In 1980, there
were only 9 AWSs, of which five were located in the Ross Island Vicinity, two stations on the coast of
Adélie Land, and two in inland Antarctica (Byrd and Dome C). The number and spatial coverage of AWSs
when their data are available peak in 2014, with a total of 146 AWSs. Approximately 90% of the AWSs
were distributed in coastal areas and regions of lower elevation. Among them, the densest regions covered
by AWSs are the Ross Ice Shelf and Victoria Land, accounting for approximately 50% of AWSs in 2014.
The gradually improved AWS network has helped fill the wide gaps in climate observations across the
whole Antarctic continent.
    Despite the significant improvement of the spatial coverage of AWSs, the data availability are still not
regularly distributed and are clustered in specific areas of Antarctica (see Fig.6, Table S5). Air
temperature and pressure are relatively easy to measure, have the highest data availability of any sensor,
and have high integrity and wide spatial coverage. Additionally, the quality of air temperature data is the
best, with only two stations missing air temperature measurement records. Measuring wind speed and
direction is a huge challenge in Antarctica, however, due to covering such a wide range of speeds from
calm/breeze to sustained hurricane intensity. A more challenge is that wind sensors freezing/breaking due
to environmental conditions (snow/riming, high winds, etc.). The loss of wind speed and direction data
mainly occur in the coastal areas of the Lambert Glacier Basin, Wilkes Land, Victoria Land, Mary Byrd
Land and Ellsworth Land. The humidity sensors may lose measurement accuracy at very cold
temperatures, and their data loss is highest. In addition to the West AIS and near the South Pole, there are
many AWSs that lack humidity measurements all year round in other parts of Antarctica.

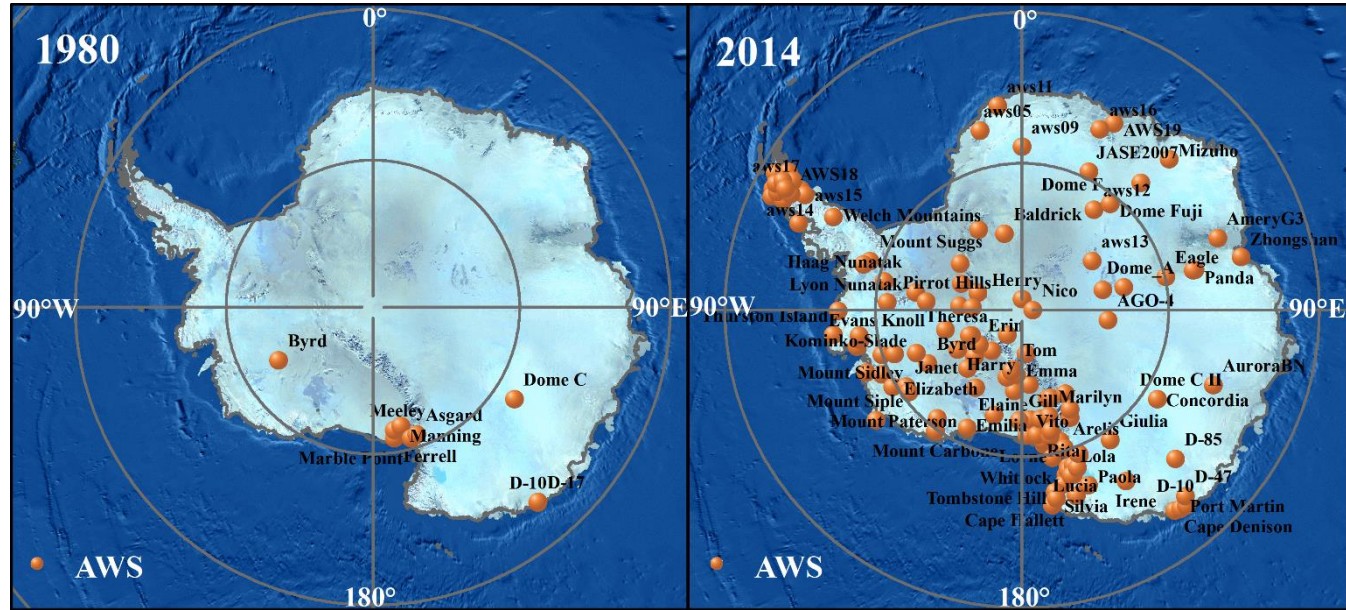

Fig.7. Spatial distribution of AWS in 1980 and 2014

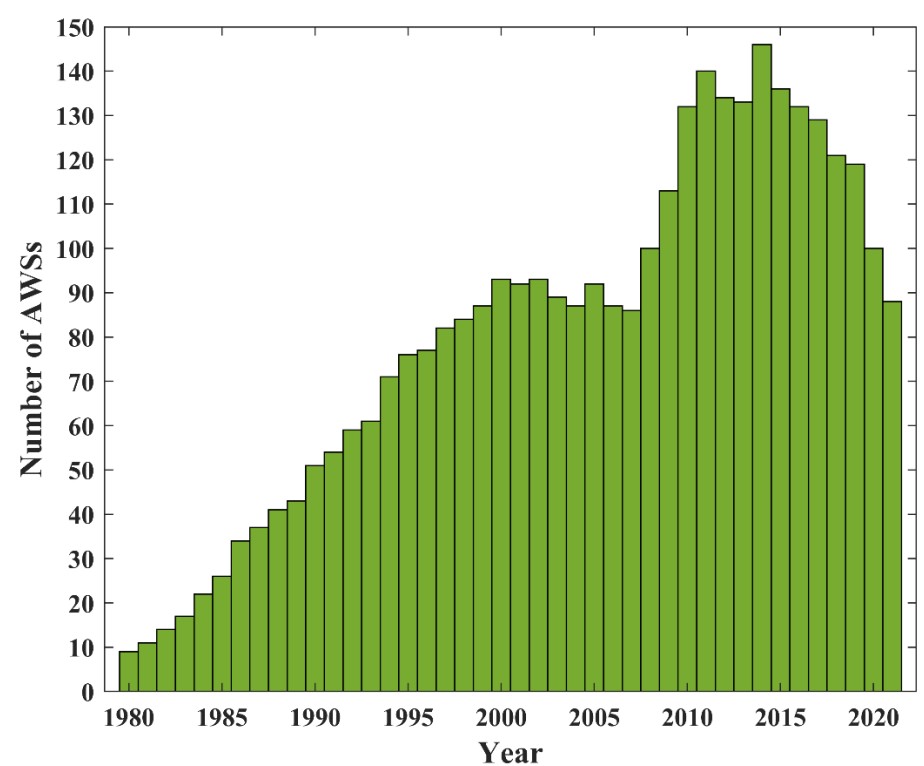

Fig.8. Number of AWSs counted each year.

## 5.2 Temporal variability in the AWS records

The five meteorological elements of each AWS cover different time spans, from 1 year to 42 years. The time period covered is closely related to sensor technology and weather conditions. Statistical results in supplemental Table S5 show that the time span of 63 AWSs exceeds 20 years, of which 27 stations exceed 30 years, but approximately 24.3% of the AWSs still cover less than 5 years. For various reasons, most of the time series in the dataset have gaps for one or all of the meteorological elements (Fig.9).

Fig.9 and Fig.S1-S4 show data availability of the daily air temperature, air pressure, wind speed and relative humidity, respectively, calculated by more than 25% of the 3-hourly observations. Among the 267 AWSs, the air temperature measurement data have the best continuity and highest data integrity. Approximately 30% of the stations have more than 15 years of daily temperature measurement data. Furthermore, 237 stations have daily data integrity of more than 50%. In recent years, the improvement of air pressure sensor technology has greatly enhanced the quality of air pressure measurement data. The integrity of daily pressure data of 225 meteorological stations exceeds 50%, and approximately 28% of stations have daily pressure data over a 15-year timespan. The wind sensor is obviously affected by temperature, and the resulting data has the poorest continuity of data. Only approximately 28% of the stations have the daily scalar wind speed and vector direction data with a timespan of more than 15 years. There are 114 stations having daily scalar wind speed and vector wind direction data integrity of more than 50%. For the 1980-2021 period, the lack of relative humidity data is the most serious, with 46 stations having no relative humidity data all year round, and only 167 stations with daily data integrity of more than 50%. Moreover, the data continuity is the lowest, with only 20% of stations measuring daily relative humidity covering more than 15 years.

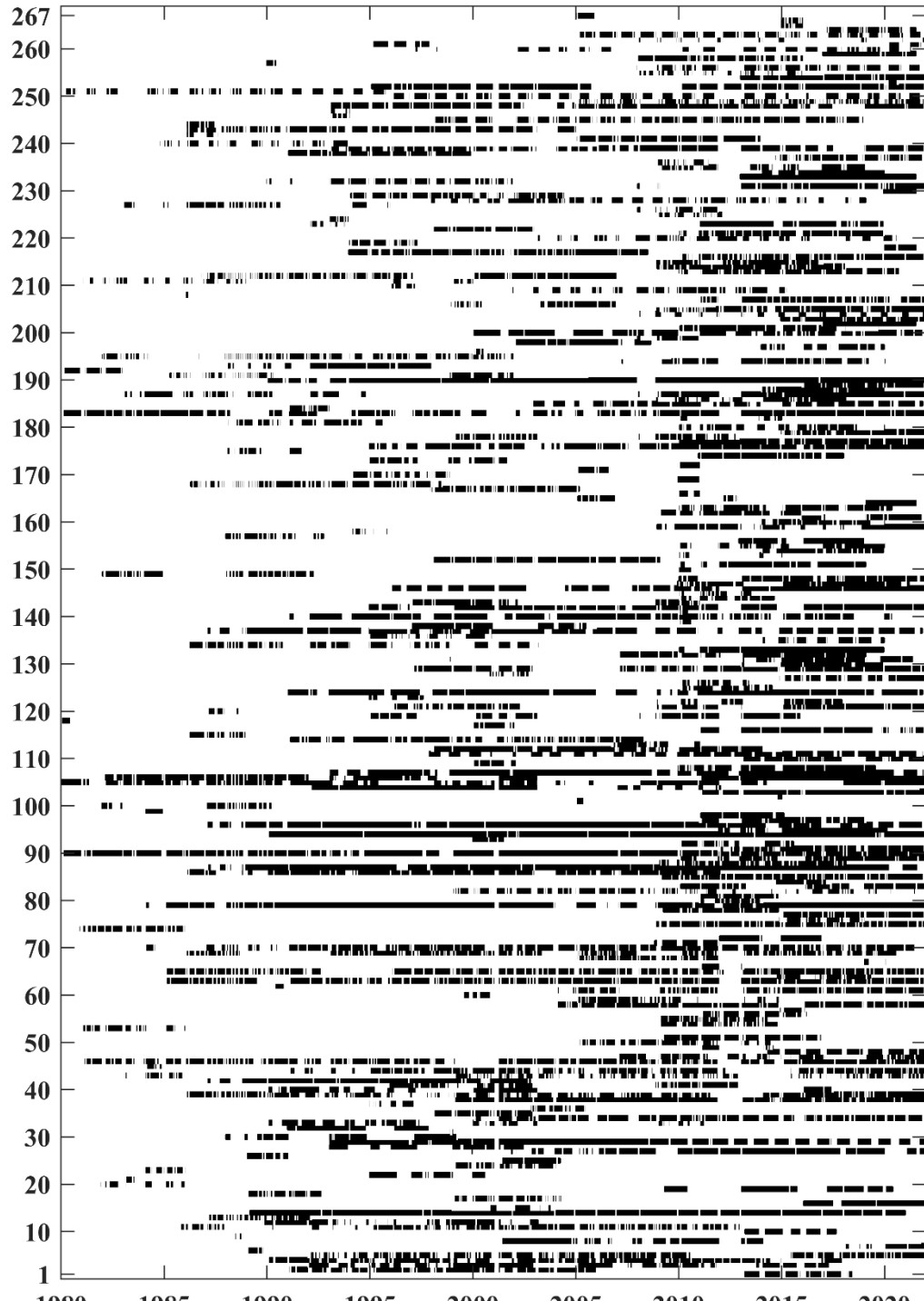

Fig. 9. Daily data availability of air temperature. Missing values have no colour, and 1-267 corresponds
to the NO. in Table S1.

**6 Station documentation**

The entire dataset consists of four subdatasets, including three quality-controlled subdatasets and one flagged subdataset of suspicious data in raw data, which are all provided in spreadsheet form. In quality-controlled daily and monthly subdatasets, all "wt" columns are the proportion of observations entered into the average value of the day or month. Number "1" indicates integrated continuous data without missing data. In the flagged subdataset, " flag_*" marks the suspicious data of each variable detected in Section 3.2 quality control. Number "4" indicates that the observed value exceeds the three standard deviations from the mean. A multiple of 100 represents the physically unrealistic 6-h rapid synoptic variability in the parameters. The air temperature records in the austral summer months (December-January-February) during the low-wind speed conditions (less than 2 m s$^{-1}$) are flagged with number "10000". Time is in 3-hourly, daily and monthly formats, and the UTC time is used in the 3-hourly data files (UTC+8). At the same time, we also provide the data integrity of 3-hourly, daily and monthly data of each variable.

The raw data we collected from different Antarctic AWS projects include four different data storage formats: ASCII format (.dat), NetCDF format (.nc), TXT format (.txt) and Excel format (.xlsx). Five meteorological elements are extracted and saved in comma separated values format (.csv format). CSV format is selected due to its simple file structure and storage mode, basic security, and extensive support in scientific applications, which is convenient for programming software (e.g., R) to process data in batches. The file names are composed using the station's name and data type. A file name such as AGO Site_3 h.csv can be read as station AGO Site, 3-hourly data, extension indicating CSV format data (.csv). The data are arranged in columns of Year, Month, Day, Three-hourly observation time (UTC), Temperature (℃), Pressure (hPa), Wind Speed (m/s), Wind Direction (°), Relative Humidity (%).

**7 Data and code availability**

The comprehensive AWS dataset is freely available as 3-hourly, daily, and monthly data separated for each station at https://amrdcdata.ssec.wisc.edu/dataset/antaws-dataset (Wang et al., 2022). All codes for the AWS data quality control are developed in the *R* environment, which are available from the corresponding authors on a reasonable request.

**8 Conclusions**

We provide a comprehensive compilation of long-term measurements of the Antarctic AWSs. The dataset includes the locations, instruments used, and measurements of five parameters i.e., air temperature, air pressure, relative humidity, wind speed and wind direction, of 267 AWSs at 3-hourly, daily and monthly resolutions, covering most areas of the Antarctic continent from 1980 to 2021. Relative to earlier studies, our compilation presents better spatial coverage, although the spatial density is least over the East Antarctic Plateau.

We adopt a comprehensive quality control process to carefully check the data to maximize the reliability of the data. This results in the reduction in the temporal density of data in some AWSs. However, the statistical results of 267 AWSs from 1980 to 2021 show that the integrity of the 3-hourly air

temperature and air pressure data from 192 stations exceeds 50%. Moreover, 159 stations have 3-hourly relative humidity data integrity of more than 50%, which is the variable of lowest data integrity. There are 92 stations with the integrity of the 3-hourly wind measurement data of less than 50%. This is easily understood as among the five variables, wind speed and direction observations have highest uncertainties caused by excessive speed, snow build-up, and so on.

The dataset can provide more accurate and effective input and verification data for the validation of reanalyses, remote sensing products and regional climate models. At the same time, as done by Steig et al. (2009), by combining the dataset with reanalysis data or remote sensing products, gridded data products can be reconstructed, which can better display the temporal and spatial variation in the AIS meteorological elements at different scales, and provide basic data for the studies of Antarctic mass balance and climate changes. It is hoped that the dataset will facilitate glaciological, meteorological, hydrological, or other studies over Antarctica.

The AWS network in the Antarctic is still incomplete and needs to be improved. In the future, it is hopeful that more AWSs will be deployed on the East Antarctic Plateau as a priority, especially on the summit of this region. However, it is highly challenging to install and maintain them in the extreme environment of the East Antarctic Plateau. Moreover, ultrasonic sounders are systematically implemented, to provide snow height data along with the meteorological data. Mechanically ventilated aspirated radiation shields should be considered to reduce radiation bias, especially in summer when solar power is available. In addition, the relative humidity supersaturated observation systems under extreme cold conditions described by Genthon et al. (2017) and Genthon et al. (2022) can be widely applied. With the continuous improvement of the AWS network and updating of AWS data, we will further refine the dataset, adopt more rigorous quality control criteria, check the unrecognizable errors in the raw data, and even provide quality marks for the dataset.

**Author contributions.**

YW contributed the idea of this work and constructed the AntAWS dataset. XZ prepared the figures and tables based on the compiled data analysis. WN wrote the codes of data processing algorithm. MAL, MD, CHR, PCJPS, PG and ERT provided part of AWS observations for constructing the dataset. MAL and PG provided some necessary information of AWSs. ZZ and YS performed the primary data collections. SH supervised this work. XZ and YW wrote the original draft, which was improved by all other authors.

**Competing interests.**

All authors have declared that none of them have any conflicts of interest.

**Acknowledgements**

Funding this work was the National Natural Science Foundation of China (41971081, 41830644 and 42122047), the National Key Research and Development Program of China (2020YFA0608202), the Strategic Priority Research Program of the Chinese Academy of Sciences (XDA19070103), the Project for Outstanding Youth Innovation Team in the Universities of Shandong Province (2019KJH011) and the Basic Research Fund of the Chinese Academy of Meteorological Sciences (2021Y021 and 2021Z006). This work is also supported by funding to the University of Wisconsin-Madison and Madison Area Technical College from the US National Science Foundation Office of Polar Programs (1924730, 1951720, and 1951603).

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
