# Peer review of "The AntAWS dataset: a compilation of Antarctic automatic weather station observations"

_Earth System Science Data, 2022_

## Referee Comment (RC2)

**Review of The AntAWS dataset: a compilation of Antarctic automatic weather station observations by Wang et al., 2022**

Wang and al. present a dataset of compiled AWS data over the Antarctic Ice Sheet. Data include near-surface temperature, humidity, wind speed and pressure. Quality checks have been performed on the data to remove outliers. In general, the original data set (3h) was already directly accessible in open access (https://amrc.ssec.wisc.edu/data/ftp/pub/aws/antrdr/) with for some already remarks on the quality of the measurements. The addition here then consists in a more thorough treatment of the reliability of the data.

**Major comment**

I have already used the original raw dataset to evaluate climate models (see remark further about the introduction) and create a compiled dataset. The quality controls I made were only visual when the comparison with both RACMO and MAR (often-used regional climate models) revealed strong disagreement with the data. If nothing looked wrong, I concluded that it was simply the models that were wrong. However, this simple method allowed me to detect many outliers and remove data while giving greater confidence in the observations. Therefore, a better outlier evaluation technique applied to these data could allow to build a very useful dataset. This is what I expected from the data. I didn't take the time to double check every data, but only a few stations for which outliers seemed to be present when I firstly used these data. I then did a quick comparison with the latest MAR results.

These values do not seem to have been removed in the AntAWS dataset. Here are some examples:
Zoraida, after 2007 the pressure decreases which seems unrealistic.

[Figure]

For instance, while RCM like MAR represent very well the pressure (eg., Motram et al., 2021, Kittel et al., 2021; Kittel 2021), the temporal correlation is very bad for the whole series (r=0.55). If I cross-check before 2007, the statistics become better (r>0.9).

Similarly, Erin and Emilia's measurements of surface pressure does not seem reliable which spurious trends.

[Figure]

I refer to Kittel, 2021 Appendix A, Table A.1 (https://orbi.uliege.be/handle/2268/258491) for the list of AWS I found.

I strongly recommend the authors to visually inspect each time series of each data before considering any publication of this database even after their statistical check. I hope that combination of several methods (statistically, physically-based methods from Wang et al., with crossed comparisons with models) would improve the reliability of the dataset. I would also suggest the authors to rewrite their introduction P1L94-96, as the same dataset has been already checked, compiled and used in several studies (eg., Mottram et al., 2021; Kittel et al., 2021; Kittel, 2021; Donat-Magnin et al., 2020; Wille et al., 2021). Consider to only insist on the availability of quality-controlled data?

**Minor comments**

It is hard to find the station location. People, when downloading the data, don't start with checking the supplement. I'd suggest to add each station location directly in the files, as well as a file with all the locations that can be directly downloaded. Section 6: L394-L395: Unless I'm mistaken, I only found the .csv files in the download link.

Section 3.3 L237-245: 25% of data availability seems really low. What is the impact of different threshold (this could be tested with correlation and rmse between the 25%dataset and X%dataset). Turner et al., 2004 used 90% (rmse of 0.1%). What is the reliability of a monthly value based on only 25% of a month? In the worst case you presented, the monthly mean value would only represent the ~first week. It is much better to have fewer reliable values than a lot of non-consistent values.

Section 4.3 L286 – 297 : Is the relatively humidity corrected for negative temperature? According to Amory (2020), the thermo-hygrometers are calibrated to measure relative humidity with respect to liquid water. Goff and Gratch (1945) formulae should then be used to convert it with respect to ice for temperature below 0°C.

**Specific remarks**

P1L29: replace estimating by evaluating

P1L35: impacts

P1L100-101: Consider to document while /where you flagged and removed some data

L137-139: 1cm is low considering the presence of moving sastrugi. Furthermore, strong temperature inversions have been found over the Antarctic Plateau (Genthon et al., 2013) which highlights the importance of this parameter.

Fig 3: What are the numbers on the map?(I guess the id of the station, but this is not mentioned in the caption)

Fig6: Why are AWS from permanent research stations like Amundsen-Scott, Dumont d'Urville, Vostok, Halley, Mc Murdo, …) not included in the data set? This strongly misleads the idea of Antarctic coverage in terms of weather stations. Furthermore, one could argue than permanent staffed stations could give more reliable data as people can check the instruments more frequently. These data could then be a significant contribution to the dataset.

Fig 8: Why do they authors use a rainbow color map?

If authors would like, I would be happy to share MAR outputs to help with outlier scan.

Sincerely,
C. Kittel

Amory, C.: Drifting-snow statistics from multiple-year autonomous measurements in Adélie Land, East Antarctica, The Cryosphere, 14, 1713–1725, https://doi.org/10.5194/tc-14-1713-2020, 2020.

Donat-Magnin, M., Jourdain, N. C., Gallée, H., Amory, C., Kittel, C., Fettweis, X., Wille, J. D., Favier, V., Drira, A., and Agosta, C.: Interannual variability of summer surface mass balance and surface melting in the Amundsen sector, West Antarctica, The Cryosphere, 14, 229–249, https://doi.org/10.5194/tc-14-229-2020, 2020.

Genthon, C., Six, D., Gallée, H., Grigioni, P., and Pellegrini, A.: Two years of atmospheric boundary layer observations on a 45-m tower at Dome C on the Antarctic plateau, J. Geophys. Res. Atmos., 118, 3218– 3232, doi:10.1002/jgrd.50128, 2013.

Goff, J. A. and Gratch, S.: Thermodynamic properties of moist air, Trans. ASHVE, 51, 125, 1945

Kittel, C.: Present and future sensitivity of the Antarctic surface mass balance to oceanic and atmospheric forcings: insights with the regional climate model MAR, PhD thesis, University of Liège, Liège, http://hdl.handle.net/2268/258491 (last access: 28 May 2022), 2021

Kittel, C., Amory, C., Agosta, C., Jourdain, N. C., Hofer, S., Delhasse, A., Doutreloup, S., Huot, P.-V., Lang, C., Fichefet, T., and Fettweis, X.: Diverging future surface mass balance between the Antarctic ice shelves and grounded ice sheet, The Cryosphere, 15, 1215–1236, https://doi.org/10.5194/tc-15-1215-2021, 2021.

Mottram, R., Hansen, N., Kittel, C., van Wessem, J. M., Agosta, C., Amory, C., Boberg, F., van de Berg, W. J., Fettweis, X., Gossart, A., van Lipzig, N. P. M., van Meijgaard, E., Orr, A., Phillips, T., Webster, S., Simonsen, S. B., and Souverijns, N.: What is the surface mass balance of Antarctica? An intercomparison of regional climate model estimates, The Cryosphere, 15, 3751–3784, https://doi.org/10.5194/tc-15-3751-2021, 2021.

Wille, J.D., Favier, V., Jourdain, N.C. et al. Intense atmospheric rivers can weaken ice shelf stability at the Antarctic Peninsula. Commun Earth Environ 3, 90. https://doi.org/10.1038/s43247-022-00422-9. 2022

---

## Community Comment (CC1)

This is a very useful paper, bringing disparate Antarctic AWS data sources together to one accessible point. I think that it should be published, although I do recommend some changes and improvements as follows.

**General comments:**

1    There is another paper also in discussion in ESSD at the moment that provides more details of one of the Antarctic AWS networks included in this compilation.  That paper is ESSD-2022-188, "The PANDA automatic weather station network from coast to Dome A, East Antarctica", Ding et al.  If both papers are accepted for publication, then it would be useful if they referenced each other.

2    In the tables and graphs, the AWS are ordered firstly by the deploying institution and then alphabetically by name.  It would be much more logical if they were sorted geographically, for example by elevation (from 0 to 4000+ m).

3    I can see no rationale for plotting the data availability Figure 8 and Figures S1-S4 in rainbow colours.  They would be simpler and clearer if they were just black and white.

4    If there are missing values in Tables S2, S3 and S4 they are reported as NaN (not a number). They should just be shown blank.

5    Maximum and minimum wind directions (in Table S2, S3, S4) are physically meaningless concepts: a maximum of 360 is the same direction as a minimum of 0. Mean wind direction is also subject to calculation error: for example, the arithmetical average of a 90- and 270-degree wind is 180 degrees; but the direction could also be 0 degrees.  A more useful statistic to show would be constancy of the wind direction (defined as the ratio of the magnitude of the mean wind vector to the scalar average wind speed).

6    It would be useful if Table S5 also included a column giving the duration (in years) that each station provided data (up to the end of 2021).

7    The English language is generally reasonable although it could be improved.  Several of the authors are native English-speakers and should review the text.

Specific comments by line number

52-53        Remote AWS became practical with the introduction of the ARGOS data relay system. This is discussed later, but should be introduced here.  The relevant satellites are not in "outer" space.

59-60        The first (successful) Australian AWS in Antarctica was deployed inland of Casey, not in the Lambert Basin.  The early Australian AWS are reported in Allison, I. and Morrissy, J.V. (1983).  Automatic weather stations in Antarctica.  Australian Meteorological Magazine, 31(2),71-76.  A network inland of Casey station was deployed during the International Antarctic Glaciology Program: those are the stations discussed in the Allison et al. 1993 paper that is cited.  Details of the Lambert Glacier Basin AWS are given in Allison, I. (1998)   The surface climate of the interior of the Lambert Glacier basin: 5 years of automatic weather station data.  Annals of Glaciology 27, 515-520.

83        "Southern Ocean island stations" NOT "South Pacific island stations"

113        Define what the acronym "PNRA" actually is

118        The supporting framework for AWS instruments differ greatly between models.  They are not "mostly tripod".

137        It would be better to give a height range. There is considerable difference between stations.

159        Table 1.  The pressure range of the AAD stations is NOT 530-791hPa: this would be useless for an AWS near sea level.  The Paroscientific sensor covers a full range of atmospheric pressure, but the

structure of the data transmitted from the stations is truncated to give a shorter message. The range is set for each AWS to cover the likely pressure range at the deployment site. The 530-791hPa range applies only to Dome A (4000+ m). Similarly, under the CHINARE stations, the range 530-791hPa is also only for the Dome A station.

160      The image of the Eagle AWS (Fig 1f) is a very poor picture of a partly buried station. I can, if requested, supply a much better image of an AAD AWS (as also deployed at Eagle and Dome A).

188-194   I found this description of cooperative links very hard to understand. I think it is incorrect in several cases.

206      Figure 4. This Figure make a lot more sense if it comes after the discussion in Section 3.2, not before. (There are also minor typographical errors in this Figure).

239      What does this mean? Surely, the purpose of an AWS is to be "unattended".

249      Not all AWS use a platinum resistor temperature probe.

253-268   With the very strong surface inversions that occur over the Antarctic plateau, a small difference in sensor height (due to different AWS design or with from accumulation with time) can be very significant. It can lead to a measured temperature difference of a degree or more over 1 metre. There is also at least one error in Table S2: aws05 (at a near coastal elevation of only 150 m) has a 3-hourly minimum temperature of -87.7! A plot of temperature vs surface elevation would be more informative than the table (and would reveal any errors).

270-271   All the AAD AWS use Paroscientific digiquartz barometers

277      A plot of pressure against surface elevation would also be more informative than the tables

289-292  The AAD AWS have included humidity measurements since about 1990. The Humicap sensor calibrations need to be corrected at very low temperatures – has this been done for all data?

329      What are the white circles in Fig. 5?

360-363  It is the length of record from the site that is important, not the length of record from an individual AWS. For example, the stations named LGB00, LGB00-A, LGB00-B and LGB00-C are all at the same site, which has a total record of ~23 years. In the Australian program, a new AWS at the same site is given a different name because it has a different calibration file: other programs may retain the same name for a replacement AWS.

409      "Antarctic AWSs" NOT "AIS AWSs". Not all stations are on the ice sheet.

---

## Community Comment (CC2)

Review of The AntAWS dataset: a compilation of Antarctic automatic weather station observations
Author(s): Yetang Wang, Xueying Zhang, Wentao Ning, Matthew A. Lazzara, Minghu Ding, Carleen H. Reijmer, Paul C. J. P. Smeets, Paolo Grigioni, Elizabeth R. Thomas, Zhaosheng Zhai, Yuqi Sun, and Shugui Hou
MS No.: essd-2022-241

Christophe Genthon, Laboratoire de Météorologie Dynamique, Paris, France

This paper presents a welcome compilation and tentative unification of data from automatic weather stations (AWS) in Antarctica.

Much of meteorological Antarctica would be essentially unknown if it was not for for the data provided by networks of AWS. Such networks have been developed and deployed by different groups, such as the AMRC, the University of Utrecht, etc, with largely similar instruments and methods but little homogeneity in the way the data are quality controlled and distributed. Yetang Wang and colleague's work is timely and definitely useful. Along with the dataset itself and associated metadata, they provide some statistics of the Antarctic meteorology and climate from AWS.

Of course one can (and should?) find that there is room for improvement. I see 2 points which need to be at least reported, possibly improved to the extent that information is available that is not yet reported in the data set.

One issue is the height of the instruments above the snow surface. The authors report that the AMRC standard height is 3 m but in practice, because of snow accumulation and infrequent visits, the height is quite variable and mostly unknown. This is an issue for temperature in areas where surface based temperature inversions can be strong (high plateau in winter). This is more particularly an issue with wind speed which varies strongly with height in the first few meters above the surface, decreasing 0 at the surface even with the strongest winds a few meters above. A few AWS (an increasing number?) have snow height variations measurements using acoustic depth gauge. This is an interesting information by itself as it measures snow accumulation which relates to snow fall and other accumulation processes such as snow erosion and blowing snow. In the first place, this is an essential information to adequately exploit the wind speed information. Unfortunately, because of snow height uncertainly, the consistency of Wang et al.'s data set is unwarrented for wind speed. This should reported.

An other issue is with the humidity. Most practical humidity sensors for AWS use Vaisala's Humicap capacitive sensor. The humicap is calibrated to report the relative humidity with respect to liquid water even below 0°C. The authors report that the relative humidity is often close to 100%, which suggest that they have converted humidity with respect to liquid into relative humidity with respect to ice. They should be clear which relative humidity they report and which conversion formulas they use. Also, it is known that in Antarctica, even near the surface, the relative humidity with respect to ice often reaches well over 100%, but few sensors can measure humidity above 100% and generally not those operated on AWS. There are recent publications in ESSD reporting this and distributing corresponding data for the high antarctic plateau. This should be reported here, and thus the fact that the database is biased low with respect to humidity, at least on the antarctic plateau.

The concluding remarks could offer some recommendations as to how to improve the AWS network. Of course we want more of these but funds are limited : which regions should be prioritized for more AWS deployment? It could be recommended that snow height sensors be systematically implemented and snow height data provided along with the meteorological data. One could also recommend to use mechanically ventilated radiation shields. Of course, power for ventilation is an issue but radiation biases occur in summer when solar power is available.

Specific comments:

Line 33: these are fairly outdated references. There surely are more recent references e.g. from the more recent IPCC reports

Line 41: I beleive this is 1958. There was no IGY in 2007. There was an international polar year started in 2007 but certainly not 50 staffed stations established then.

Lines 63-64: Why are those other AWS left aside? One major virtue of the work presented here is the efforts made to collate, harmonize and consistently distribute data which are otherwise scattered here and there. Why leave aside some data known to exist?

Line 106 and further: CR1000 is a device, not a series. It is a datalogger and should be presented as such, as this is the way the manufacturer Campbell Sci presents it. Campbell Sci should show as the manufacturer.

Line 110: Verify with BAS but initially (circa 200s), BAS made their own data loggers. They shifted to CR1000 later on.

Lines 118-119: hard to understand: is this a tripod or a mast? In fact most long term AWS are on masts, e.g. AMRC's.

Line 137: This is the problem, nominal height, possibly known at deployment and after visits but most of the time it is fully unknown unless the AWS is equiped with an ADG which is generally not the case.

Lines 139-140: Sorry but this is this is a ridiculous  estimation of the error. Eisen et al. is about long term mean snow accumulation, and they report accumulations up to and more than 1 m / year in some places in Antarctica. Surely the height uncertainty issue is less where accumulation is less,

e.g. on the high plateau, but this uncertainty is first a matter of mean accumulation and servicing frequency.

Table 1: Any information here on where temperature reports may benefit aspirated radiation shielding to avoid radiation biases?

Also in table 1: I am a bit confused with the term "impeller". Vane manufacturer R. M. Young, for instance, call it "propeller"

Still Table 1, BAS is reported using HMP155 resistance probe for relative humidity. HMP155 actually uses the Humicap capacitive sensor. The temperature report from HMP155 uses platinum resistance to report temperature, not humidity.

Lines 178-179: please provide internet links for consistency with other sources of information. Otherwise, should this be "personal communication"?

Figure 4: In the data processing step, should this be "flagging" rather than interpolating?

Lines 209-211: concerning the 3-hour time step: are the data instant measures every 3 hours , or averaged over 3 hours? Is this consistent across datasets? How do you average wind direction?

Lines 232-232: did / could you check that no mechanical ventilation is used before blacklisting low wind cases? This is probably mostly the case, but should there be some valid reports by low wind speed thanks to mechanical ventilation?

Line 239: the 3-value criteria should probably also include that the 3 values are homogeneously distributed during the day, otherwise the a time-of-day bias is likely is summer when temperature strongly varies with sun elevation.

Line 252: again, any indication that some temperature reports may benefit mechanical ventilation?

Line 262: in fact, Dome C and Concordia are one and the same site, if not necessarily the same AWS. No wonder they show the same extremes. I suggest keep only Dome C here.

Section 4.3: please mention the relative humidity issues raised above here: sensors report RH with respect to liquid, data and must be converted to get RH with respect to ice; and the sensors used on AWS cannot report supersaturation, which is frequent on the high antarctic plateau – the humidity data are thus biased low there.

Section 4.4: please mention that poorly known instrument height above the snow surface affects the data quality / consistency. Still, the time evolution of wind speed with time is an important information, but the modulus is not well known and not consistent in the dataset.

Figures S1, S2, S3, S4: mention that there is no color code, colors are used to improve readability?

---

## Author Comment (AC2)

**Respond to the comments of RC1 (Changqing Ke)**

This paper provides a new quality-controlled dataset of meteorological records from Antarctic automatic weather stations (AntAWS dataset) at 3-hourly, daily and monthly resolutions. The dataset compiles the measurements of air temperature, air pressure, relative humidity, and wind speed and direction from 216 AWSs available during 1980-2021. This dataset will be valuable for better characterizing surface climatology throughout the continent of Antarctica, improving our understanding of Antarctic surface snow-atmosphere interactions, and estimating regional climate models or meteorological reanalysis products. It can be published after minor revision.

Response:

We are grateful to the reviewer for the great work and his recognition of the value on our study. We realized that he has a great expertise to make most useful comments and suggestions. We also appreciate the constructive comments and suggestion, and we have considered all the points, and please see our point-by-point responses on the comments.

1. Fig.1's resolution is too low, should be replaced with high quality pictures.

Response:

The resolution of the Fig.1 will be greatly improved using a higher resolution picture of an AAD AWS (Ian Allison have agreed to provide the picture), and other pictures. We will finish this when submitting the revised manuscript.

2. Fig.2 with same problem, very low resolution.

Response:

We have further improved its resolution.

3. Fig.3 should add some main location names.

Response:

Thank you for your constructive comments, we have added some main Antarctic location names to the Fig.3 in the revised manuscript. However, we still choose to use the form of digital annotation for the station names, because the number of sites is too large, making it is relatively messy to add the station names. The corresponding site

names of the numbers on the map are presented in Table S1. We have added this in the Fig.3 caption, as follows.

[Figure]

Fig.3. Mapping the sites of 267 Automatic Weather Stations (AWSs), the numbers (1-267) corresponds to NO. in Table S1.

4. Fig.5's caption should add full names which have abbreviations on the maps, for example, Tmax means maximum temperature, etc.

Response:

Following your advice, the corresponding changes have made in the Fig.5's caption accordingly.

5. P4 L143-144, 'snow height' should be 'snow depth'.

Response:

It has been modified.

---

## Author Comment (AC3)

**Respond to the comments of CC2 (Christophe Genthon)**

Review of The AntAWS dataset: a compilation of Antarctic automatic weather station observations Author(s): Yetang Wang, Xueying Zhang, Wentao Ning, Matthew A. Lazzara, Minghu Ding, Carleen H. Reijmer, Paul C. J. P. Smeets, Paolo Grigioni, Elizabeth R. Thomas, Zhaosheng Zhai, Yuqi Sun, and Shugui Hou

MS No.: essd-2022-241

Christophe Genthon, Laboratoire de Météorologie Dynamique, Paris, France

This paper presents a welcome compilation and tentative unification of data from automatic weather stations (AWS) in Antarctica.

Much of meteorological Antarctica would be essentially unknown if it was not for for the data provided by networks of AWS. Such networks have been developed and deployed by different groups, such as the AMRC, the University of Utrecht, etc, with largely similar instruments and methods but little homogeneity in the way the data are quality controlled and distributed. Yetang Wang and colleague's work is timely and definitely useful. Along with the dataset itself and associated metadata, they provide some statistics of the Antarctic meteorology and climate from AWS.

Of course one can (and should?) find that there is room for improvement. I see 2 points which need to be at least reported, possibly improved to the extent that information is available that is not yet reported in the data set.

Response:

   We highly thank the excellent review work that you do for our manuscript. We are also grateful to you for the recognition of our work. All your comments have been considered and the manuscript has been revised accordingly. Please see our point-by-point responses on the specific comments.

1. One issue is the height of the instruments above the snow surface. The authors report that the AMRC standard height is 3 m but in practice, because of snow accumulation and infrequent visits, the height is quite variable and mostly unknown. This is an issue

for temperature in areas where surface based temperature inversions can be strong (high plateau in winter). This is more particularly an issue with wind speed which varies strongly with height in the first few meters above the surface, decreasing 0 at the surface even with the strongest winds a few meters above. A few AWS (an increasing number?) have snow height variations measurements using acoustic depth gauge. This is an interesting information by itself as it measures snow accumulation which relates to snow fall and other accumulation processes such as snow erosion and blowing snow. In the first place, this is an essential information to adequately exploit the wind speed information. Unfortunately, because of snow height uncertainly, the consistency of Wang et al.'s data set is unwarrented for wind speed. This should reported.

Response:

The relevant descriptions have been added in the first paragraph of Section 4.1 and 4.4, and recommendations for reducing air temperature and wind observational uncertainty have been provided in Section 8, and they are as follows.

Section 4.1

"*It should be emphasized that over the areas with strong temperature inversions, especially high plateau in winter, near-surface air temperature is influenced by the changes in the height of sensors installed over the AWS (generally a relative "lowering") caused by snow accumulation (Genthon et al., 2021).*"

Section 4.4

"*It is important to recall that wind speed varies strongly with height in the first few meters above the surface, and the height of the sensors above surface gradually decreases with snow accumulation, causing poorly known variations of the instrument height above the snow surface, affects the data quality and consistency (Genthon et al., 2021). Still, the evolution of wind speed with time is an important information, but the modulus is not well known and not consistent in the dataset. To improve the accuracy of air temperature and wind observations, the vertical temperature and wind profiles should be corrected by accounting for the sensor height variations, as done by Ma et al. (2008) and Smeets et al. (2018). However, this additional computed data will be left until we have sufficient snow height data.*"

Reference:

Genthon, C., Veron, D., Vignon, E., Six, D., Dufresne, J.-L., Madeleine, J.-B., Sultan, E., and Forget, F.: 10 years of temperature and wind observation on a 45 m tower at Dome C, East Antarctic plateau, Earth Syst. Sci. Data, 13, 5731–5746, https://doi.org/10.5194/essd-13-5731-2021, 2021.

Ma, Y., Bian, L., Xiao, C., Allison, I.: Correction of snow accumulation impacted on air temperature from automatic weather station on the Antarctic Ice Sheet. Advance in Polar Science, 20: 299-309, http://ir.casnw.net/handle/362004/7877, 2008.

Smeets, P. C., Kuipers Munneke, P., Van As, D., van den Broeke, M. R., Boot, W., Oerlemans, H., Snellen, H., Reijmer, C.H., and van de Wal, R. S.: The K-transect in west Greenland: Automatic weather station data (1993-2016), Arctic, Antarctic, and Alpine Research, 50, S100002, https://doi.org/10.1080/15230430.2017.1420954, 2018.

2. An other issue is with the humidity. Most practical humidity sensors for AWS use Vaisala's Humicap capacitive sensor. The humicap is calibrated to report the relative humidity with respect to liquid water even below 0°C. The authors report that the relative humidity is often close to 100%, which suggest that they have converted humidity with respect to liquid into relative humidity with respect to ice. They should be clear which relative humidity they report and which conversion formulas they use. Also, it is known that in Antarctica, even near the surface, the relative humidity with respect to ice often reaches well over 100%, but few sensors can measure humidity above 100% and generally not those operated on AWS. There are recent publications in ESSD reporting this and distributing corresponding data for the high antarctic plateau. This should be reported here, and thus the fact that the database is biased low with respect to humidity, at least on the antarctic plateau.

Response:

We agree with you. Few sensors can measure humidity above 100%, and it is unreasonable to assume that relative humidity is often close to 100% in the Antarctic. The relative humidity is only available at this point computed with respect to liquid

water and not with respect to ice. For this reason, we have further improved the data quality control criteria and adjusted the relative humidity threshold to less than 100%. The relevant descriptions have been added in the first paragraph of section 4.3 accordingly, and recommendation for improving relative humidity measurements have been provided in the section 8, as follows.

Section 4.3

*"The Vaisala humicap, which itself takes the conversion of ice and water form into account, is factory calibrated to provide RH with respect to liquid water even at below-freezing temperatures (Amory, 2020; Genthon, et al., 2013). The relative humidity is only available at this point computed with respect to liquid water. Data should be converted to get RH with respect to ice using the method of Goff and Gratch (1945) (Amory, 2020), but this additional computed data are left for the forthcoming papers. And the sensors used on AWS cannot report supersaturation, the relative humidity with respect to ice often reaches well over 100%, in Antarctica, even near the surface, especially which is frequent on the high Antarctic plateau (Genthon, et al., 2017, 2022). Therefore, the database is biased low with respect to humidity."*

Reference:

Amory, C.: Drifting-snow statistics from multiple-year autonomous measurements in Adélie Land, East Antarctica, The Cryosphere, 14, 1713–1725, https://doi.org/10.5194/tc-14-1713-2020, 2020.

Goff, J. A. and Gratch, S.: Thermodynamic properties of moist air, Trans. ASHVE, 51, 125, 1945.

Genthon, C., Six, D., Gallée, H., Grigioni, P., and Pellegrini, A.: Two years of atmospheric boundary layer observations on a 45-m tower at Dome C on the Antarctic plateau, Journal of Geophysical Research: Atmospheres, 118, 3218–3232, https://doi.org/10.1002/jgrd.50128, 2013.

Genthon, C., Piard, L., Vignon, E., Madeleine, J.-B., Casado, M., and Gallée, H.: Atmospheric moisture supersaturation in the near-surface atmosphere at Dome C, Antarctic Plateau, Atmos. Chem. Phys., 17, 691–704, https://doi.org/10.5194/acp-17-691-2017, 2017.

Genthon, C., Veron, D. E., Vignon, E., Madeleine, J.-B., and Piard, L.: Water vapor in cold and clean atmosphere: a 3-year data set in the boundary layer of Dome C, East Antarctic Plateau, Earth Syst. Sci. Data, 14, 1571–1580, https://doi.org/10.5194/essd-14-1571-2022, 2022.

3. The concluding remarks could offer some recommendations as to how to improve the AWS network. Of course we want more of these but funds are limited: which regions should be prioritized for more AWS deployment? It could be recommended that snow height sensors be systematically implemented and snow height data provided along with the meteorological data. One could also recommend to use mechanically ventilated radiation shields. Of course, power for ventilation is an issue but radiation biases occur in summer when solar power is available.

Response:

Thank you for your constructive comments, we have added some recommendations as to how to improve the AWS network in the Conclusion. The new additions as follows.

"*However, the AWS network in the Antarctic is still incomplete and needs to be improved. In the future, it is hoped that more AWS will be deployed on the East Antarctic Plateau as a priority, especially on the summit of the East Antarctic Plateau. However, it is very challenging to install and maintain them in the extreme environment of the East Antarctic Plateau. Moreover, ultrasonic sounders are systematically implemented, to provide snow height data along with the meteorological data. And mechanically ventilated aspirated radiation shields should be considered to reduce radiation bias, especially in summer when solar power is available. In addition, the relative humidity supersaturated observation systems under extreme cold conditions described by Genthon et al. (2017) and Genthon et al. (2022) can be widely applied.*"

Specific comments:

1. Line 33: these are fairly outdated references. There surely are more recent references e.g. from the more recent IPCC reports

Response:

It has been modified. We have changed the references in the revision, as follows.

References

Intergovernmental Panel on Climate Change.: IPCC special report on the ocean and cryosphere in a changing climate, https://archive.ipcc.ch/srocc/, 2019.

Kennicutt, M. C. II, Bromwich, D., Liggett, D., Njåstad, B., Peck, L., Rintoul, S. R., Ritz, C., Siegert, M. J., Aitken, A., Brooks, C. M., Cassano, J., Chaturvedi, S., Chen, D., Dodds, K., Golledge, N. R., Bohec, C. L., Leppe, M., Murray, A., Nath, P. C., Raphael, M. N., Rogan-Finnemore, M., Schroeder, D. M., Talley, L., Travouillon, T., Vaughan, D. G., Wang, L., Weatherwax, A. T., Yang, H., Chown, S. L.: Sustained Antarctic research: a 21st century imperative, One Earth, 1, 95–113, https://doi.org/10.1016/j.oneear.2019.08.014, 2019.

Rignot, E., Mouginot, J., Scheuchl, B., and Morlighem, M.: Four decades of Antarctic Ice Sheet mass balance from 1979–2017, PNAS, 116, 1095-1103, https://doi.org/10.1073/pnas.1812883116, 2019.

2. Line 41: I beleive this is 1958. There was no IGY in 2007. There was an international polar year started in 2007 but certainly not 50 staffed stations established then.

Response:

Yes, you are right. We have modified this mistake and now is "*For example, a total of approximately 50 staffed stations were established by the end of the IGY, of which 17 have continuous meteorological records to date (Lazzara et al., 2013; Summerhayes et al., 2008).*"

3. Lines 63-64: Why are those other AWS left aside? One major virtue of the work presented here is the efforts made to collate, harmonize and consistently distribute data which are otherwise scattered here and there. Why leave aside some data known to exist?

Response:

We have corrected this misstatement and now is "*In 1985, the PNRA (the Italian National Programme of Antarctic Research) installed its first AWS "Mario Zucchelli" in Terra Nova Bay. Currently its AWS network mainly located in the region of Victoria Land and the Antarctic Plateau. Over the Antarctic Peninsula and Dronning Maud Land, the British Antarctic Survey and the Institute for Marine and Atmospheric Research, Utrecht University (IMAU) installed their respective AWS network. The*

*CHINARE (Chinese National Antarctic Research Expedition) mainly develop PANDA automatic weather station network, including eleven AWSs from the coast to the summit of the East Antarctic plateau (Ding et al., 2022). There are other AWS networks in the Antarctic that are included in this project (e.g. Japan, France, New Zealand, South Korea, etc.).*"

As you mentioned, we try to organize, coordinate and consistently distribute data, and we collected as many site data as we could, also including weather stations in Japan (e.g., Dome Fuji, Mizuho, JASE2007, etc.), France (e.g., D-47, D-85, D-17, D-10, etc.), New Zealand (e.g., Minna Bluff, etc.), Korea (e.g., Bear Peninsula, etc.), and so on.

4. Line 106 and further: CR1000 is a device, not a series. It is a datalogger and should be presented as such, as this is the way the manufacturer Campbell Sci presents it. Campbell Sci should show as the manufacturer.

Response:

We have modified CR1000 series or system to CR1000 device and showed Campbell Sci as the manufacturer in the manuscript.

5. Line 110: Verify with BAS but initially (circa 200s), BAS made their own data loggers. They shifted to CR1000 later on.

Response:

It has been modified in the revision and now is "*Initially, AWSs created by the British Antarctic Survey (BAS) use their own data loggers, and then switch to use the CR1000 device for measurements.*"

6. Lines 118-119: hard to understand: is this a tripod or a mast? In fact most long term AWS are on masts, e.g. AMRC's.

Response:

Thank you for pointing out the problems and the sentence has been corrected as "*The supporting framework for AWS instruments varies between models. But in general, the AWS body is made up of a mast, usually with instrument arms fitted with different sensors.*"

7. Line 137: This is the problem, nominal height, possibly known at deployment and after visits but most of the time it is fully unknown unless the AWS is equiped with an

ADG which is generally not the case.

Response:

Yes, you are right. It has been modified and now is "*Each AWS measures air temperature, pressure, relative humidity and other meteorological elements within an initial height range of 1~4 m and/or 6 m above the ground (reference to the initial height from build stations, snow accumulation and site tilt were not part of the monitored variables), except for Zhongshan Station, which measures wind speed and wind direction at a height of 10 meters.*"

Due to lack of snow accumulation measurement data, we don't correct the air temperature and wind speed by considering the height changes of sensors. If the snow accumulation observations are available, we will do the corresponding corrections in the future.

8. Lines 139-140: Sorry but this is a ridiculous estimation of the error. Eisen et al. is about long term mean snow accumulation, and they report accumulations up to and more than 1 m / year in some places in Antarctica. Surely the height uncertainty issue is less where accumulation is less, e.g. on the high plateau, but this uncertainty is first a matter of mean accumulation and servicing frequency.

Response:

Sorry for this mistake. Following your advice, we have deleted this ridiculous estimation of the error and made the following modifications.

"*Due to the accumulation of snow, the measurement height of each meteorological element varies over time, which may result in the notable meteorological measurement disparities such as temperature and wind speed due to instrument height differences.*"

9. Table 1: Any information here on where temperature reports may benefit aspirated radiation shielding to avoid radiation biases?

Response:

We have checked that air temperature and relative humidity we collected are measured inside naturally ventilated non-aspirated radiation shields. Energy considerations do not allow mechanically aspirated shields of the temperature/humidity sensors.

Regarding AMRC AWS, there only have a couple test cases of using aspiration (at Henry and Nico), but we don't have the results of those. Nothing has been published about the Henry/Nico AWS data, but it definitely showed notable warming in non-aspirated vs aspirated shields well outside the error of the sensor. For the aspirated shield, van den Broeke (2005) goes out of the way to find a correction based on incoming solar radiation and wind speed to correct for radiation errors during low wind. And regarding impact of low wind speed on radiation bias and how aspirated shields would correct this, we can refer to Genthon et al. (2011). However, this does not belong to the purpose of this paper. In the future, we can continue to improve in the future research.

Reference:

Genthon, C., Six, D., Favier, V., Lazzeri, M., and Keller, L.: Atmospheric temperature measurement biases on the Antarctic plateau, Journal of Atmospheric and Oceanic Technology, 28, 1598–1605, https://doi.org/10.1175/JTECH-D-11-00095.1, 2011.

Van den Broeke, M.: Strong surface melting preceded collapse of Antarctic Peninsula ice shelf, Geophysical Research Letters, 32, L12815, https://doi.org/10.1029/2005GL023247, 2005.

10. Also in table 1: I am a bit confused with the term "impeller". Vane manufacturer R. M. Young, for instance, call it "propeller"

Response:

We are very sorry for this mistake, "impeller" has been modified to "propeller" in the revision.

11. Still Table 1, BAS is reported using HMP155 resistance probe for relative humidity. HMP155 actually uses the Humicap capacitive sensor. The temperature report from HMP155 uses platinum resistance to report temperature, not humidity.

Response:

It has been modified in Table 1.

12. Lines 178-179: please provide internet links for consistency with other sources of information. Otherwise, should this be "personal communication"?

Response:

The internet link has been added, as follows.

"*the Chinese National Antarctic Research Expedition (CHINARE) (https://doi.org/10.11888/Atmos.tpdc.272721).*"

13. Figure 4: In the data processing step, should this be "flagging" rather than interpolating?

Response:

We have redrawn the Figure 4 and modified it, as follows.

[Figure]

Fig.4. Description of AWSs data processing process.

14. Lines 209-211: concerning the 3-hour time step: are the data instant measures every

3 hours, or averaged over 3 hours? Is this consistent across datasets? How do you average wind direction?

Response:

The 3-hour time step means data selected at three hourly intervals, which a three-hour data generation method based on the AMRC. Our compilation is based on the hourly and 3-hourly synoptic measurements from AWSs, data selected at three hourly intervals, produce a three hourly data set for each AWS. Due to the inconsistent time steps of the collected datasets, we adopted this method to unify the data into three-hour data in order to unify the data structure.

When doing the averaging for wind, we break the wind speed and direction into components, and computed the resultant wind. For daily and monthly data, we used arithmetic average method and vector average method to calculate scalar and vector wind speed and direction, respectively.

15. Lines 232-232: did/could you check that no mechanical ventilation is used before blacklisting low wind cases? This is probably mostly the case, but should there be some valid reports by low wind speed thanks to mechanical ventilation?

Response:

Yes, we have check that no mechanical ventilation is used before blacklisting low wind cases. Air temperature we collect is measured inside naturally ventilated radiation shields. There only have a couple test cases (at Henry and Nico), but we don't have the results of those. There generally haven't mechanically aspirated shields due to power budget, and fan failure resulting in significantly worse data results.

16. Line 239: the 3-value criteria should probably also include that the 3 values are homogeneously distributed during the day, otherwise the a time-of-day bias is likely is summer when temperature strongly varies with sun elevation.

Response:

In the revision, we have added the 75% to calculate the daily and monthly dataset, that is, at least six 3-hourly observed values are available, referring to Kittel (2021). This ensures the distribution during the day as much as possible and minimizes data errors.

We will be glad to modify again, if change didn't follow your intention.

17. Line 252: again, any indication that some temperature reports may benefit mechanical ventilation?

Response:

Air temperature data we collected are measured inside naturally ventilated radiation shields mainly because of limited energy resources and the logistical access required to operate and maintain ventilation. There only have a couple test cases (at Henry and Nico), but we don't have the results of those.

18. Line 262: in fact, Dome C and Concordia are one and the same site, if not necessarily the same AWS. No wonder they show the same extremes. I suggest keep only Dome C here.

Response:

Concordia has been deleted, and we keep only Dome C here.

19. Section 4.3: please mention the relative humidity issues raised above here: sensors report RH with respect to liquid, data and must be converted to get RH with respect to ice; and the sensors used on AWS cannot report supersaturation, which is frequent on the high antarctic plateau – the humidity data are thus biased low there.

Response:

As the relative humidity issue raised above, we have added the corresponding description of this important issue in Section 4.3, as detailed above.

20. Section 4.4: please mention that poorly known instrument height above the snow surface affects the data quality/ consistency. Still, the time evolution of wind speed with time is an important information, but the modulus is not well known and not consistent in the dataset.

Response:

As the wind issue raised above, we have added a corresponding description of this important issue in Section 4.4, as detailed above.

21. Figures S1, S2, S3, S4: mention that there is no color code, colors are used to improve readability

Response:

Yes, you are right. We ignored the color code in the preliminary analysis. Rainbow color map are used to improve readability, but based on the feedback, it didn't work. Therefore, in the revision, we have changed the rainbow colors of Figure 8 and Figures S1-S4 to black and white for simplicity and clarity.

---

## Author Comment (AC4)

Review of The AntAWS dataset: a compilation of Antarctic automatic weather station observations by Wang et al., 2022

Wang and al. present a dataset of compiled AWS data over the Antarctic Ice Sheet. Data include near-surface temperature, humidity, wind speed and pressure. Quality checks have been performed on the data to remove outliers. In general, the original data set (3h) was already directly accessible in open access (https://amrc.ssec.wisc.edu/data/ftp/pub/aws/antrdr/) with for some already remarks on the quality of the measurements. The addition here then consists in a more thorough treatment of the reliability of the data.

Response:

   We would like to thank Christoph Kittel (the reviewer) for doing the thorough review, and for the thoughtful and constructive comments and suggestions that improved the quality of our manuscript. All your comments have been considered and the manuscript has been revised accordingly. Please see our point-by-point responses on the major and specific comments.

**Major comment**

1. I have already used the original raw dataset to evaluate climate models (see remark further about the introduction) and create a compiled dataset. The quality controls I made were only visual when the comparison with both RACMO and MAR (often-used regional climate models) revealed strong disagreement with the data. If nothing looked wrong, I concluded that it was simply the models that were wrong. However, this simple method allowed me to detect many outliers and remove data while giving greater confidence in the observations. Therefore, a better outlier evaluation technique applied to these data could allow to build a very useful dataset. This is what I expected from the data. I didn't take the time to double check every data, but only a few stations for which outliers seemed to be present when I firstly used these data. I then did a quick comparison with the latest MAR results.

These values do not seem to have been removed in the AntAWS dataset. Here are some

examples:

Zoraida, after 2007 the pressure decreases which seems unrealistic.

[Figure]

For instance, while RCM like MAR represent very well the pressure (eg., Motram et al., 2021, Kittel et al., 2021; Kittel 2021), the temporal correlation is very bad for the whole series (r=0.55). If I cross-check before 2007, the statistics become better (r>0.9). Similarly, Erin and Emilia's measurements of surface pressure does not seem reliable which spurious trends.

[Figure]

[Figure]

I refer to Kittel, 2021 Appendix A, Table A.1 (https://orbi.uliege.be/handle/2268/258491) for the list of AWS I found.

I strongly recommend the authors to visually inspect each time series of each data before considering any publication of this database even after their statistical check. I hope that combination of several methods (statistically, physically-based methods from Wang et al., with crossed comparisons with models) would improve the reliability of the dataset.

Response:

   Thank you for your constructive comments. After our quality control, we have visually inspected each time series of each data, and cross-compared with ERA5 to remove outliers. When the MAR outputs are available, we will visually check and perform cross-comparison with MAR data.

   In addition, we are investigating the error of Emilia's pressure change – there is an error, and we are working on finding out what happened, and when, and if we cannot find out a why, we will explore fixing it if we can or if we can't fix it, at least remove what is determined to be bad data. This investigation will not be complete in time for this revision of our manuscript. This investigation was underway before it was pointed out by this reviewer.

   We are not clear on why Erin AWS has errors in its pressure measurement. We thank the reviewer for pointing this out and we will launch an investigation into that. The

investigation will not be complete before the revision of this manuscript is complete.

I would also suggest the authors to rewrite their introduction P1L94-96, as the same dataset has been already checked, compiled and used in several studies (eg., Mottram et al., 2021; Kittel et al., 2021; Kittel, 2021; Donat-Magnin et al., 2020; Wille et al., 2021). Consider to only insist on the availability of quality-controlled data?

Response:

We have rewritten the introduction P1L94-96 and now is "*These AWS observation compilations generally suffer from part or all of the following limitations: the duration of datasets, single meteorological parameter, low temporal resolution, limited spatial coverage, no rigorous quality control and in some cases limited availability for the public. Most recently, Kittel compiled a near-surface weather observation database at a high temporal resolution, which to a great extent remedied the deficiency of the previous database (Kittel, 2021), and has already been used in the studies of the ice sheet surface processes, climate model validation or atmospheric diagnoses (e.g., Donat-Magnin et al., 2020; Mottram et al., 2021; Kittel et al., 2021; Kittel, 2021; Wille et al., 2021). However, this database were only visual crossed comparisons with models to detect many outliers and then remove them, and it is still not available for the public. Thus, better quality control could allow to build a more reliable dataset.*"

References:

Donat-Magnin, M., Jourdain, N. C., Gallée, H., Amory, C., Kittel, C., Fettweis, X., Wille, J. D., Favier, V., Drira, A., and Agosta, C.: Interannual variability of summer surface mass balance and surface melting in the Amundsen sector, West Antarctica, The Cryosphere, 14, 229–249, https://doi.org/10.5194/tc-14-229-2020, 2020.

Kittel, C.: Present and future sensitivity of the Antarctic surface mass balance to oceanic and atmospheric forcings: insights with the regional climate model MAR, PhD thesis, University of Liège, Liège, http://hdl.handle.net/2268/258491 (last access: 28 May 2022), 2021

Kittel, C., Amory, C., Agosta, C., Jourdain, N. C., Hofer, S., Delhasse, A., Doutreloup, S., Huot, P.-V., Lang, C., Fichefet, T., and Fettweis, X.: Diverging future surface mass balance between the Antarctic ice shelves and grounded ice sheet, The

Cryosphere, 15, 1215–1236, https://doi.org/10.5194/tc-15-1215-2021, 2021.

Mottram, R., Hansen, N., Kittel, C., van Wessem, J. M., Agosta, C., Amory, C., Boberg, F., van de Berg, W. J., Fettweis, X., Gossart, A., van Lipzig, N. P. M., van Meijgaard, E., Orr, A., Phillips, T., Webster, S., Simonsen, S. B., and Souverijns, N.: What is the surface mass balance of Antarctica? An intercomparison of regional climate model estimates, The Cryosphere, 15, 3751–3784, https://doi.org/10.5194/tc-15-3751-2021, 2021.

Wille, J.D., Favier, V., Jourdain, N.C. et al. Intense atmospheric rivers can weaken ice shelf stability at the Antarctic Peninsula. Commun Earth Environ 3, 90. https://doi.org/10.1038/s43247-022-00422-9. 2022

**Minor comments**

1. It is hard to find the station location. People, when downloading the data, don't start with checking the supplement. I'd suggest to add each station location directly in the files, as well as a file with all the locations that can be directly downloaded. Section 6: L394-L395: Unless I'm mistaken, I only found the .csv files in the download link.

Response:

Thanks for your constructive comments, we have added all the station locations that can be directly downloaded. However, in order to make it easier for users to batch process the data using programming software, we don't add each station location directly in the data product separately. If you insist, we will be glad to added it in each file.

You're not mistaken. The raw data we collected from different Antarctic AWS project databases are stored in different data forms, and we have unified them into CSV files. In the download link, we only provide our dataset (.csv format). It has been modified and now is "*The raw data we collected from different Antarctic AWS project include four different data storage formats: ASCII format (.dat), NetCDF format (.nc), TXT format (.txt) and Excel format (.xlsx).*"

2. Section 3.3 L237-245: 25% of data availability seems really low. What is the impact of different threshold (this could be tested with correlation and rmse between the

25%dataset and X%dataset). Turner et al., 2004 used 90% (rmse of 0.1%). What is the reliability of a monthly value based on only 25% of a month? In the worst case you presented, the monthly mean value would only represent the ~first week. It is much better to have fewer reliable values than a lot of non-consistent values.

Response:

    Referring to the daily and monthly data processing method of the AMRC, we used a threshold of 25%, in order to provide as much data as possible. Considering the reliability of the data, we also provide daily data and monthly data products calculated using a 75% threshold, that is, at least six 3-hourly observed values are available, referring to Kittel (2021). This ensures the distribution during the day as much as possible and minimizes data errors.

    We will be glad to modify again, if change didn't follow your intention.

3. Section 4.3 L286 – 297: Is the relatively humidity corrected for negative temperature? According to Amory (2020), the thermo-hygrometers are calibrated to measure relative humidity with respect to liquid water. Goff and Gratch (1945) formulae should then be used to convert it with respect to ice for temperature below 0°C.

Response:

    We didn't consider the corrections of the RH data at the negative temperature. Most practical humidity sensors for AWS use Vaisala's Humicap capacitive sensor. The Vaisala humicap, which itself takes the conversion of ice and water form into account, is factory calibrated to provide RH with respect to liquid water even at below-freezing temperatures (Genthon et al., 2013). The relative humidity is only available at this point computed with respect to liquid water and not with respect to ice. We appreciate the interest and hope to accomplish this additional computed data value in the future, but not before this manuscript is ready for resubmission.

    The relevant descriptions have been added to "4.3 Relative humidity".

Reference:

Genthon, C., Six, D., Gallée, H., Grigioni, P., and Pellegrini, A.: Two years of atmospheric boundary layer observations on a 45-m tower at Dome C on the Antarctic plateau, Journal of Geophysical Research: Atmospheres, 118, 3218–3232,

https://doi.org/10.1002/jgrd.50128, 2013.

**Specific remarks**

1. P1L29: replace estimating by evaluating

Response:

   Done.

2. P1L35: impacts

Response:

   It has been modified in the revision.

3. P1L100-101: Consider to document while /where you flagged and removed some data

Response:

   Thanks for your constructive comments, we have generated one flagged subdataset of suspicious data in the raw dataset. See Section 6 for detailed flag instructions.

4. L137-139: 1cm is low considering the presence of moving sastrugi. Furthermore, strong temperature inversions have been found over the Antarctic Plateau (Genthon et al., 2013)

which highlights the importance of this parameter.

Response:

   Thank you for pointing out the problems, and we fully agree with you.

   Here our attempts are just to discuss the impact of sensor height changes due to snow accumulation on the meteorological measurements, not to discuss the uncertainty of snow height observations. Sorry for this mistake, and the corresponding corrections have been made, and they are as follow.

   "*Due to the accumulation of snow, the measurement height of each meteorological element varies over time, which may result in the notable meteorological measurement disparities such as temperature and wind speed due to instrument height differences.*"

5. Fig 3: What are the numbers on the map? (I guess the id of the station, but this is not mentioned in the caption)

Response:

The numbers (1-267) on the map correspond to NO. in Table S1. We have added this in the Fig 3 caption, as follows.

[Figure]

Fig.3. Mapping the sites of 267 Automatic Weather Stations (AWSs), the numbers (1-267) corresponds to NO. in Table S1.

6. Fig6: Why are AWS from permanent research stations like Amundsen-Scott, Dumont d'Urville, Vostok, Halley, Mc Murdo, …) not included in the data set? This strongly misleads the idea of Antarctic coverage in terms of weather stations. Furthermore, one could argue than permanent staffed stations could give more reliable data as people can check the instruments more frequently. These data could then be a significant contribution to the dataset.

Response:

We have added the AWSs from the POLENET program. In addition, this paper was about AWS– non-staffed stations. It was not about Vostok or McMurdo or South Pole that are staffed fully or staffed at least part of the year with people making or managing observational equipment. If the point of the work is to be complete of all surface observations, that will change things. A statement has been added to denote that this

work was not focused on staffed stations.

7. Fig 8: Why do they authors use a rainbow color map?

Response:

Rainbow color map are used to improve readability, but based on the feedback, it didn't work. Therefore, in the revision, we have changed the rainbow colors of Figure 8 and Figures S1-S4 to black and white for simplicity and clarity.

8. If authors would like, I would be happy to share MAR outputs to help with outlier scan.

Response:

Thank you very much. Indeed, we really need MAR outputs to help with the outlier scan and further improve the reliability of the dataset.

---

## Author Comment (AC6)

**Respond to the comments of CC1 (Ian Allison)**

This is a very useful paper, bringing disparate Antarctic AWS data sources together to one accessible point. I think that it should be published, although I do recommend some changes and improvements as follows.

Response:

We would like to thank you for doing this review, and for the useful comments and suggestions that improved our manuscript. All your comments have been considered and the manuscript has been revised accordingly. Please see our point-by-point responses on the specific comments.

General comments:

1. There is another paper also in discussion in ESSD at the moment that provides more details of one of the Antarctic AWS networks included in this compilation. That paper is ESSD-2022-188, "The PANDA automatic weather station network from coast to Dome A, East Antarctica", Ding et al. If both papers are accepted for publication, then it would be useful if they referenced each other.

Response:

Thanks for your good advice, we have referenced the paper ESSD-2022-188 in the introduction, section 2 and section 4.4 of the revised version.

Reference:

Ding, M., Zou, X., Sun, Q., Yang, D., Zhang, W., Bian, L., Lu, C., Allison, I., Heil, P., and Xiao, C.: The PANDA automatic weather station network between the coast and Dome A, East Antarctica, Earth Syst. Sci. Data Discuss. [preprint], https://doi.org/10.5194/essd-2022-188, in review, 2022.

2. In the tables and graphs, the AWS are ordered firstly by the deploying institution and then alphabetically by name. It would be much more logical if they were sorted geographically, for example by elevation (from 0 to 4000+ m).

Response:

Followed your advices, we have modified the tables and graphs and sorted them according to station elevation changes, as follows.

[Figure]

Fig.3. Mapping the sites of 267 Automatic Weather Stations (AWSs), the numbers (1-267) corresponds to NO. in Table S1.

3. I can see no rationale for plotting the data availability Figure 8 and Figures S1-S4 in rainbow colours. They would be simpler and clearer if they were just black and white.
Response:

We have changed the rainbow colors of Figure 8 and Figures S1-S4 to black and white. We will finish this when submitting the revised manuscript.

4. If there are missing values in Tables S2, S3 and S4 they are reported as NaN (not a number). They should just be shown blank.
Response:

The missing values in Table S2, S3 and S4 have been changed as blank.

5. Maximum and minimum wind directions (in Table S2, S3, S4) are physically meaningless concepts: a maximum of 360 is the same direction as a minimum of 0. Mean wind direction is also subject to calculation error: for example, the arithmetical average of a 90- and 270-degree wind is 180 degrees; but the direction could also be 0 degrees. A more useful statistic to show would be constancy of the wind direction

(defined as the ratio of the magnitude of the mean wind vector to the scalar average wind speed).

Response:

Thanks for your instructive suggestions. In revision, we have deleted the maximum and minimum wind directions, and changed the mean wind direction to the constancy of the wind direction in Tables S2, S3 and S4.

6. It would be useful if Table S5 also included a column giving the duration (in years) that each station provided data (up to the end of 2021).

Response:

Followed your advices, in Table S5, we have added a column giving the duration (in years) that each station provided data (up to the end of 2021).

7. The English language is generally reasonable although it could be improved. Several of the authors are native English-speakers and should review the text.

Response:

The English has been improved by Matthew A. Lazzara, Elizabeth R. Thomas, David Mikolajczyk, Lee J. Welhouse, and Linda M. Keller. Hopefully, the revised version is readable.

Specific comments by line number

1. 52-53 Remote AWS became practical with the introduction of the ARGOS data relay system. This is discussed later, but should be introduced here. The relevant satellites are not in "outer" space.

Response:

Following your advice, corresponding changes have been made accordingly, and now it is "*Remote AWS became practical with the introduction of the ARGOS data relay system on polar orbiting satellites in 1978, and thus real-time or near real-time meteorological data could be obtained in distant places.*"

2. 59-60 The first (successful) Australian AWS in Antarctica was deployed inland of Casey, not in the Lambert Basin. The early Australian AWS are reported in Allison, I. and Morrissy, J.V. (1983). Automatic weather stations in Antarctica. Australian Meteorological Magazine, 31(2),71-76. A network inland of Casey station was

deployed during the International Antarctic Glaciology Program: those are the stations discussed in the Allison et al. 1993 paper that is cited. Details of the Lambert Glacier Basin AWS are given in Allison, I. (1998) The surface climate of the interior of the Lambert Glacier basin: 5 years of automatic weather station data. Annals of Glaciology 27, 515-520.

Response:

Thank you for pointing out the problems in this section and providing the important reference. Corresponding changes have been made accordingly, and now it is "*In 1982, the AAD deployed its first AWS in Antarctic inland Casey (*Allison and Morrissy, 1983*). During the International Antarctic Glaciology Program, a network inland of Casey station was deployed (Allison et al., 1993). Later, the National Antarctic Research Expedition (ANARE) of Australia set up an AWS network with updated version of the stations in the Lambert Glacier drain of East Antarctica (Allison et al., 1998).*"

References:

Allison, I., and Morrissy, J.V.: Automatic weather stations in Antarctica, Australian Meteorological Magazine, 31, 71-76, 1983.

Allison, I.: The surface climate of the interior of the Lambert Glacier basin: 5 years of automatic weather station data, Annals of Glaciology, 27, 515-520, https://doi:10.3189/1998AoG27-1-515-520, 1998.

3. 83 "Southern Ocean island stations" NOT "South Pacific island stations"

Response:

It has been modified.

4. 113 Define what the acronym "PNRA" actually is

Response:

PNRA: the Italian National Programme of Antarctic Research. The introduce has been modified and added to the location where it firstly appeared.

5. 118 The supporting framework for AWS instruments differ greatly between models. They are not "mostly tripod".

Response:

Thank you for pointing out the problems and we have corrected the sentence as "*The supporting framework for AWS instruments varies between models. But in general, the*

*AWS body is made up of a mast, usually with instrument arms fitted with different sensors."*

6. 137 It would be better to give a height range. There is considerable difference between stations.

Response:

It has been modified and now it is "*Each AWS measures air temperature, pressure, relative humidity and other meteorological elements within an initial height range of 1~4 m and/or 6 m above the ground (reference to the initial height from build stations, snow accumulation and site tilt were not part of the monitored variables), except for Zhongshan Station, which measures wind speed and wind direction at a height of 10 m.*"

7. 159 Table 1. The pressure range of the AAD stations is NOT 530-791hPa: this would be useless for an AWS near sea level. The Paroscientific sensor covers a full range of atmospheric pressure, but the structure of the data transmitted from the stations is truncated to give a shorter message. The range is set for each AWS to cover the likely pressure range at the deployment site. The 530-791hPa range applies only to Dome A (4000+ m). Similarly, under the CHINARE stations, the range 530-791hPa is also only for the Dome A station.

Response:

They have been modified in Table 1. The Paroscientific Digiquartz 6015A covers 0 to 1100 hPa; The ranges set for the Dome A, Eagle, LGB69 AWSs are 530 to 610 hPa, 635 to 735 hPa, and 691 to 791 hPa, respectively.

8. 160 The image of the Eagle AWS (Fig 1f) is a very poor picture of a partly buried station. I can, if requested, supply a much better image of an AAD AWS (as also deployed at Eagle and Dome A).

Response:

Thank you for providing the picture. We have used one to replace the image of the Eagle AWS (Fig 1f), as follows.

[Figure]

Fig.1. Typical AWSs of the six research institutions, but the sensors at other sites vary slightly depending on the local environment. a) AMRC-CR1000 device, b) AMRC-AGO-4, c) AMRC and CHINARE-Panda_South, d) IMAU-AWS10, e) PNRA-Maria, f) AAD-LGB00, g) BAS-the sensors used on Latady, h) BAS-Latady.

a) http://amrc.ssec.wisc.edu/news/2010-May-01.html
b) https://amrc.ssec.wisc.edu/aws/images/station_images/AGO_4.jpg
c) personal communication with Minghu Ding.
d) https://www.projects.science.uu.nl/iceclimate/aws/technical.php
e) https://www.climantartide.it/attivita/aws/index.php?lang=en
f) personal communication with Ian Allison
g) and h) https://ramadda.data.bas.ac.uk/repository/entry/show?entryid=synth%3A44d1a477-0852-4620-a1f4-63f559b44e94%3AL0RvY3VtZW50cy9waG90b3NfYXdz

9. 188-194 I found this description of cooperative links very hard to understand. I think it is incorrect in several cases.

Response:

Yes, you are right. The AMRC brings together data from several Antarctic research programs, not cooperative links. It has been modified and now it is "*The AMRC includes not only its own AWSs network but also brings together data from several Antarctic research programs, such as the Japanese Antarctic Research Expedition (JARE), the French Antarctic Program (Institut Polaire Francais-Paul Emile Victor, IPEV), the AAD, the BAS and the CHINARE. The JARE installed and maintained the JASE2007, Dome Fuji, Mizuho and the Relay Station on the East Antarctic Plateau. The IPEV installed and took charge of the AWSs from the Adélie Coast to Dome C II, including the Port Martin, D-10, D-17, D-47, D-85, Dome C and Dome C II. The cape Denison on the Adélie Coast belongs to AAD. The BAS installed and maintained the AWSs on the AP and the East Antarctic Plateau, including the Butler Island, Larsen Ice Shelf, Limbert, Sky-Blu, Fossil Bluff, Dismal Island and the Baldrick. The Panda South station, located on the East Antarctic Plateau, a cooperation between CHINARE and AMRC, which was installed, maintained and operated by CHINARE.*"

10. 206 Figure 4. This Figure make a lot more sense if it comes after the discussion in Section 3.2, not before. (There are also minor typographical errors in this Figure).

Response:

We have redrawn Figure 4 and put it after the discussion in Section 3.2, as follows.

[Figure]

Fig.4. Description of AWSs data processing process.

11. 239 What does this mean? Surely, the purpose of an AWS is to be "unattended".

Response:

It means an AWS infrequent visits. It has been modified and now it is "*Unfortunately, a number of events that occur may result in data gaps because of only checked periodically.*"

12. 249 Not all AWS use a platinum resistor temperature probe.

Response:

It has been modified and now it is "*Air temperature is a sensitive indicator of the climate extremes experienced by the whole continent, which is measured at heights of*

*approximately 3 m above the ground based on the thermistor (such as Apogee ST-110 Thermistor and FS23D thermistor in ratiometric circuit) or resistive platinum probe (such as PRT series and Vaisala HMP series).*"

13. 253-268 With the very strong surface inversions that occur over the Antarctic plateau, a small difference in sensor height (due to different AWS design or with from accumulation with time) can be very significant. It can lead to a measured temperature difference of a degree or more over 1 metre. There is also at least one error in Table S2: aws05 (at a near coastal elevation of only 150 m) has a 3-hourly minimum temperature of -87.7! A plot of temperature vs surface elevation would be more informative than the table (and would reveal any errors).

Response:

We agree with you. Especially due to snow accumulation, uncertainties of measuring temperatures occur in areas with the very strong surface inversions (high plateau in winter). The relevant descriptions have been added in the first paragraph of Section 4.1, and recommendations for reducing air temperature measurement uncertainties have been provided in Section 8, and they are as follows.

Section 4.1

"*It should be emphasized that over the areas with strong temperature inversions, especially high plateau in winter, near-surface air temperature is influenced by the changes in the height of sensors installed over the AWS (generally a relative "lowering") caused by snow accumulation (Genthon et al., 2021).*"

Section 8

"*However, the AWS network in the Antarctic is still incomplete and needs to be improved. In the future, it is hoped that more AWS will be deployed on the East Antarctic Plateau as a priority, especially on the summit of the East Antarctic Plateau. However, it is very challenging to install and maintain them in the extreme environment of the East Antarctic Plateau. Moreover, ultrasonic sounders are systematically implemented, to provide snow height data along with the meteorological data. And mechanically ventilated aspirated radiation shields should be considered to reduce radiation bias, especially in summer when solar power is available. In addition, the relative humidity supersaturated observation systems under extreme cold conditions described by Genthon et al. (2017) and Genthon et al. (2022) can be widely applied.*"

And, we have corrected the errors in Table S2, and visually inspected each time series

of each data and cross-compared with ERA5 to remove the outliers.

A plot of temperature against surface elevation chart has been added in the manuscript.

Reference:

Genthon, C., Piard, L., Vignon, E., Madeleine, J.-B., Casado, M., and Gallée, H.: Atmospheric moisture supersaturation in the near-surface atmosphere at Dome C, Antarctic Plateau, Atmos. Chem. Phys., 17, 691–704, https://doi.org/10.5194/acp-17-691-2017, 2017.

Genthon, C., Veron, D., Vignon, E., Six, D., Dufresne, J.-L., Madeleine, J.-B., Sultan, E., and Forget, F.: 10 years of temperature and wind observation on a 45 m tower at Dome C, East Antarctic plateau, Earth Syst. Sci. Data, 13, 5731–5746, https://doi.org/10.5194/essd-13-5731-2021, 2021.

Genthon, C., Veron, D. E., Vignon, E., Madeleine, J.-B., and Piard, L.: Water vapor in cold and clean atmosphere: a 3-year data set in the boundary layer of Dome C, East Antarctic Plateau, Earth Syst. Sci. Data, 14, 1571–1580, https://doi.org/10.5194/essd-14-1571-2022, 2022.

14. 270-271 All the AAD AWS use Paroscientific digiquartz barometers

Response:

It has been modified and now it is "*All the AAD AWSs use Paroscientific digiquartz barometers, with an accuracy of ±0.2 hPa and a resolution of 0.1 hPa. AMRC AWSs also use Paroscientific digiquartz barometers (Paroscientific Model 215 A), which have an higher resolution of 0.04 hPa and accuracy of ±0.1 hPa.*"

15. 277 A plot of pressure against surface elevation would also be more informative than the tables

Response:

A plot of pressure against surface elevation chart has been added in the manuscript.

16. 289-292 The AAD AWS have included humidity measurements since about 1990. The Humicap sensor calibrations need to be corrected at very low temperatures – has this been done for all data?

Response:

Firstly, we have modified this mistake. And we don't consider the corrections of the

RH data at very low temperatures. Most practical humidity sensors for AWS use Vaisala's Humicap capacitive sensor. The Vaisala humicap, which itself accounts for the conversion of ice and water form, is factory calibrated to provide RH with respect to liquid water even at below-freezing temperatures (Genthon et al., 2013). The relative humidity is only available at this point computed with respect to liquid water and not with respect to ice. We appreciate the interest and hope to accomplish this additional computed data value in the future, but not before this manuscript is ready for resubmission.

The relevant description has been added to "4.3 Relative humidity".

The detail information with humidity and temperature probe, please see: https://www.vaisala.com/en/products/weather-environmental-sensors/humicap-humidity-temperature-probe-hmp155.

https://www.vaisala.com/sites/default/files/documents/HMP45AD-User-Guide-U274EN.pdf.

https://www.vaisala.com/sites/default/files/documents/WEA-MET-ProductSpotlight-HMP155-B212226EN-A.pdf.

Reference:

Genthon, C., Six, D., Gallée, H., Grigioni, P., and Pellegrini, A.: Two years of atmospheric boundary layer observations on a 45-m tower at Dome C on the Antarctic plateau, Journal of Geophysical Research: Atmospheres, 118, 3218–3232, https://doi.org/10.1002/jgrd.50128, 2013.

17. 329 What are the white circles in Fig. 5?

Response:

White circles represent missing data. We have added the description in the caption of Fig.5.

18. 360-363 It is the length of record from the site that is important, not the length of record from an individual AWS. For example, the stations named LGB00, LGB00-A, LGB00-B and LGB00-C are all at the same site, which has a total record of ~23 years. In the Australian program, a new AWS at the same site is given a different name

because it has a different calibration file: other programs may retain the same name for a replacement AWS.

Response:

Following your advice, we have intergraded records of AAD stations at the same nine sites, and then calculated their respective total record length. The nine sites are ① A028, A028-A and A028-B, ②DSS and DSS-A, ③GF08 and GF08-A, ④Lanyon and Lanyon-A, ⑤LawDome and LawDome-A, ⑥LGB00, LGB00-A, LGB00-B and LGB00-C, ⑦LGB10 and LGB10-A, ⑧LGB69 and LGB69-A, and ⑨MtBrown and MtBrown-A.

19. 409 "Antarctic AWSs" NOT "AIS AWSs". Not all stations are on the ice sheet.

Response:

It has been modified in the manuscript.

---

## Author Response (AR1)

We enclose below the comments from the editor, reviewers and scientific community. Comments are addressed in the same order as in the reviews. The comments are in black fonts and our responses are in blue fonts. Petra Heil, David Mikolajczyk, Lee J. Welhouse, and Linda M. Keller have significantly contributed to the quality of this revised work with fruitful discussions and English improvements. They are therefore co-authorship in the current version.

**Response to the editor**

We await reviewer comments but, for this reader, saving data by station (200-some) produces too many files? Some creative compilation might prove useful. All or none download not the best option?

Response:

To easy download, we have updated the dataset as a compressed file and added each station location directly in a subfolder.

**Response to the comments of reviewer one (Changqing Ke)**

This paper provides a new quality-controlled dataset of meteorological records from Antarctic automatic weather stations (AntAWS dataset) at 3-hourly, daily and monthly resolutions. The dataset compiles the measurements of air temperature, air pressure, relative humidity, and wind speed and direction from 216 AWSs available during 1980-2021. This dataset will be valuable for better characterizing surface climatology throughout the continent of Antarctica, improving our understanding of Antarctic surface snow-atmosphere interactions, and estimating regional climate models or meteorological reanalysis products. It can be published after minor revision.

**Response:**

We are grateful to the reviewer for the great work and his recognition of the value on our study. We realized that he has a great expertise to make most useful comments and suggestions. We also appreciate the constructive comments and suggestion, and we have considered all the points, and please see our point-by-point responses on the comments. 1. Fig.1's resolution is too low, should be replaced with high quality pictures.

Response:

The resolution of the Fig.1 will be greatly improved using a higher resolution picture of an AAD AWS (Ian Allison have agreed to provide the picture), and other pictures. We will finish this when submitting the revised manuscript.

2. Fig.2 with same problem, very low resolution.

Response:

We have further improved its resolution.

3. Fig.3 should add some main location names.

Response:

Thank you for your constructive comments, we have added some main Antarctic location names to the Fig.3 in the revised manuscript. However, we still choose to use the form of digital annotation for the station names, because the number of sites is too large, making it is relatively messy to add the station names. The corresponding site names of the numbers on the map are presented in Table S1. We have added this in the Fig.3 caption, as follows.

---

## Author Response (AR2)

**Response to the editor**

Something odd at lines 130-131. A few other typos. Authors can fix all remaining errors at proof stage.

Response:

We have rephrased the sentence, and other editing corrections have been made throughout the text.